# Position: AI for Science Should Treat Measurement-to-Dataset Pipelines as Inference Components

Ling Zhan [* 1 2]   Xiaoyao Yu [* 1]   Tao Jia [1 2 3]

## Abstract

AI for Science (AI4Science) workflows often treat the released dataset as a fixed interface to the underlying system. However, in domains relying on *indirect observation*, the learner observes a derivative representation produced by multi-stage measurement, reconstruction, and preprocessing pipelines. **We argue that these measurement-to-dataset pipelines are inference components: treating their outputs as "given data" freezes an observation model and obscures uncertainty over feasible pipeline choices.** We identify three failure modes arising from this "frozen lens": **(C1) hidden hypothesis space**, where the released dataset does not specify the pipeline configuration or its validity conditions; **(C2) uncertified transportability**, where a pipeline may be documented but its regime of validity is untested, so failures under distribution shift cannot be adjudicated; **(C3) ungoverned multiplicity**, where many defensible pipelines exist and dispersion is real but not propagated into uncertainty-aware evidence. We stress-test these claims with a large-scale neuroscience empirical audit, finding a survival rate of $\approx 0.0004\%$ under a cross-dataset stability criterion. We call on the AI4Science community to make pipelines *computable* inference objects via domain-specific Computable Observation Frameworks. This shift enables quantifying pipeline adequacy and stability, converting implicit implementation choices into auditable, reproducible, and cumulative scientific evidence.

---

[*]Equal contribution [1] College of Computer and Information Science, Southwest University, Chongqing, China [2] Chongqing Key Laboratory of Brain-Inspired Cognitive Computing and Educational Rehabilitation for Children with Special Needs, Chongqing Normal University, Chongqing, China [3] College of Computer and Information Science, Chongqing Normal University, Chongqing, China. Correspondence to: Tao Jia <tjia@swu.edu.cn>.

*Proceedings of the 43rd International Conference on Machine Learning*, Seoul, South Korea. PMLR 306, 2026. Copyright 2026 by the author(s).

## 1. Introduction

AI for Science (AI4Science) is often framed around benchmark-driven predictive objectives, yet the epistemic aims of science are broader: hypothesis generation, theory building, and mechanistic explanation tied to the underlying system (Wang et al., 2023; Messeri & Crockett, 2024). A growing literature integrates modern machine learning (ML) into discovery workflows across disciplines, from multi-messenger astrophysics to computational psychology (Xu et al., 2021; Van Noorden & Perkel, 2023; Cuoco et al., 2022; Ke et al., 2025). However, in many fields, research findings are mined from *indirect observation*: the inference of a scientific target from proxy measurements via a nontrivial observation model, effectively treating data acquisition as an inverse problem (Stuart, 2010; Murata et al., 2023). For example, in radio interferometry, images are reconstructed from visibilities via calibration and deconvolution choices (Thiébaut, 2013). The same issue also appears in structural biology and drug discovery, where downstream models often consume curated structural or bioactivity records whose apparent "data" status depends on deposition, assay, normalization, and filtering conventions (Gaulton et al., 2017). **Appendix A** gives representative pipelines spanning five indirect-observation domains.

Consequently, AI4Science under indirect observation must consider not only *what* to learn, but also *which pipeline configuration* defines the representation to be learned from. This is a difference of degree rather than kind. Standard ML also depends on preprocessing, but its inferential object usually centers on the mapping from representation to prediction on an agreed task interface. Preprocessing typically serves this objective as performance-oriented engineering (Zha et al., 2025; Wang et al., 2026). In indirect-observation AI4Science, configuration $\phi$ specifies a multi-stage mapping from sensor measurements to a released representation, with consequential degrees of freedom that are rarely specified, constrained, or audited (Hand, 2004; Ahmed, 2025). Because AI4Science often aims to support *mechanistic* claims about an underlying system that is not directly observable, the configured pipeline functions as an *observation model* linking measurements to hypothesized latent structure. Downstream learning can optimize per-

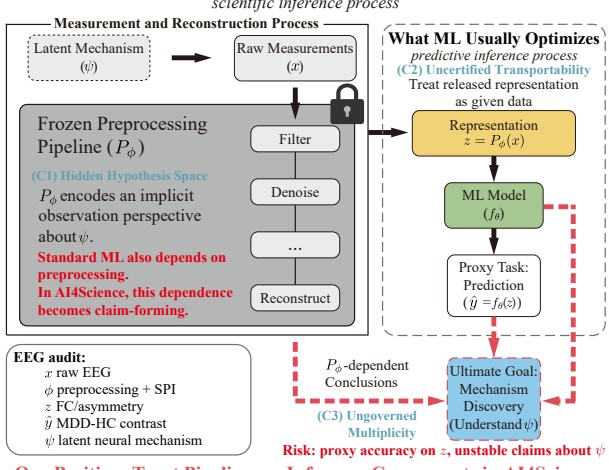

*Figure 1.* Predictive inference versus scientific inference. The contrast is a difference of degree rather than kind: standard ML also depends on preprocessing. In indirect-observation AI4Science, however, a pipeline $P_\phi$ transforms raw measurements $x$ into a released representation $z = P_\phi(x)$, from which a downstream model produces $\hat{y} = f_\theta(z)$. Mechanistic claims about the latent target $\psi$ are therefore conditional on $\phi$. The inset gives the EEG audit instantiation.

formance on the released interface, but it yields evidence for the claimed mechanism only conditional on $\phi$ (Arridge et al., 2019; Antun et al., 2020).

*Figure* 1 summarizes this shift from predictive inference to scientific inference. Both settings may instantiate the same computational template, but when the target moves from prediction to mechanism discovery, $P_\phi$ directly shapes how mechanisms are inferred and interpreted. In indirect-observation AI4Science, released representations depend on weakly identified defaults, inherited recipes, and tuning decisions that encode assumptions about signal and artifact (Botvinik-Nezer et al., 2020; Breznau et al., 2022; Kiar et al., 2024). Freezing these choices yields a *frozen lens*: an implicit inference component whose output is treated as an objective data fact (Paullada et al., 2021; Leonelli, 2019). This can reward predictive proxies while hiding pipeline uncertainty from the learner, weakening mechanistic interpretation (Markowetz, 2024).

**Our position: Pipelines are components of inference in AI4Science. They should be audited and evaluated under scientific validity criteria, not frozen as dataset definitions. Uncertainty over admissible observation models should be quantified and carried forward as cumulative evidence, rather than collapsed to a point choice.**

We articulate this position through three failure modes. **(C1)** Hidden hypothesis space: the released representation hides the pipeline-indexed space of mechanisms that could have

been tested. **(C2)** Uncertified transportability: a documented pipeline rarely comes with a tested regime of validity across measurement settings. **(C3)** Ungoverned multiplicity: multiple defensible pipelines remain, but their dispersion is rarely propagated into uncertainty-qualified evidence.

The paper proceeds by characterizing how pipelines become dataset interfaces (Section 2), deriving three failure modes (Section 3; Appendix B), auditing EEG connectomics under admissible pipeline variation (Section 4; Appendix C), addressing common objections (Section 5), and proposing computable observation frameworks (Section 6).

## 2. The Frozen Lens in AI4Science

This section characterizes how current AI4Science workflows turn observation pipelines into dataset interfaces. Across pipeline-first, differentiable, and foundation-model paradigms, what travels across labs is often a derivative representation rather than raw measurement. *Table 1* summarizes how each paradigm fixes, learns, or inherits a pipeline configuration $\phi$ before downstream learning begins.

### 2.1. The Evolution of the Frozen Lens

In the common AI4Science regime, the data–model interface is shaped by modularity. Domain practitioners in computational chemistry, astrophysics, and neuroimaging apply established toolchains (e.g., RDKit, Astropy, MNE-Python) to raw measurements or primary observational products to produce model-ready representations (Hirsch & Gilad-Bachrach, 2021; Klarner et al., 2023; Caucheteux et al., 2021). This separation is practical and reinforced by processed-derivative benchmarks, where the community distributes the released representation as the evaluation object, thereby fixing $\phi$ upstream (Glasser et al., 2013; Baptista et al., 2022). As a result, the dependence of derivative datasets on pipeline choices is often weakly identified from the available data and correspondingly under-reported (Kessler et al., 2025; Merz et al., 2023).

The frozen-lens problem is not created by AI. Pre-AI scientific workflows already used toolchains, reporting norms, quality-control checklists, and shared derivative formats to stabilize analysis across laboratories (Gorgolewski et al., 2016; Esteban et al., 2019; Nichols et al., 2017). These mechanisms improve reproducibility, but they usually document the pipeline that was run rather than the admissible family of pipelines that could have been run. AI4Science amplifies the consequence of this arrangement: processed derivatives become benchmark targets, learned front-ends are optimized and then frozen, and foundation models inherit preprocessing and reconstruction conventions through their training corpora (Strypsteen & Bertrand, 2021; Bushuiev et al., 2025; Sun et al., 2025). As reuse

*Table 1.* **Bundling pipeline into "data" in current AI4Science workflows.** Community-facing interfaces typically fix an observation model via a pipeline $P_\phi$ (explicit, inherited, or learned-then-frozen). $\phi$ indexes pipeline choices.

| Paradigm | Typical Instantiations | Practical Interface | Role / Status of $\phi$ |
|---|---|---|---|
| **Pipeline-first** (classical) | Processed-derivative benchmarks (Glasser et al., 2013; Baptista et al., 2022). Standardized toolchains (Landrum, 2013; Robitaille et al., 2013; Gramfort et al., 2014). | A released representation (often with a reference pipeline/QC recipe) becomes the object of reuse and benchmarking. | *Explicit and fixed:* $\phi$ is selected upstream and executed once to produce the representation. |
| **Hybrid / Differentiable** (end-to-end) | Learnable front-ends (Strypsteen & Bertrand, 2021); Differentiable operators (Engel et al., 2020); Unrolled physics-based modules (Aggarwal et al., 2019). | A deployed end-to-end system (front-end + learner/reconstructor) with a standard inference API. | *Learned then fixed:* $\phi$ is optimized during training, then deployed as a fixed front-end. |
| **Model-first** (foundation models) | Curated pretraining data with fixed reconstruction/harmonization conventions (Bushuiev et al., 2025; Sun et al., 2025; Soares et al., 2025). | A pretrained model and its representation space (embeddings, generative priors, or task adapters). | *Implicit and inherited:* $\phi$ is encoded in pretraining curation and reconstruction conventions. |

and automation scale, a fixed $\phi$ can propagate farther from the original measurement context and be optimized against more aggressively by downstream learners (Paullada et al., 2021; Sambasivan et al., 2021a; Thiyagalingam et al., 2022).

Hybrid approaches make parts of the pipeline differentiable and learnable, turning preprocessing into trainable modules (Engel et al., 2020; Aggarwal et al., 2019). This can learn $\phi$ end-to-end, but the learned front-end is usually optimized for predictive or reconstruction loss and then deployed as a fixed interface. Sensitivity of scientific conclusions to the learned pipeline is therefore often left to post hoc checks rather than treated as a governed commitment (Antun et al., 2020).

Scientific foundation models shift, rather than remove, the frozen lens dynamic (Nguyen et al., 2023; Pyzer-Knapp et al., 2025). When trained on raw measurements, the observation model is absorbed into the training distribution and model parameters, making $P_\phi$ harder to audit (Bushuiev et al., 2025; Szwarcman et al., 2025). When trained on released representations, $\phi$ is inherited from corpus-level reconstruction, filtering, and curation conventions (Sun et al., 2025; Soares et al., 2025). In both regimes, prediction may improve while the pipeline shaping the representation becomes harder to inspect, vary, or validate (Bommasani et al., 2025; Sambasivan et al., 2021b).

## 2.2. Systematic Constraints and Epistemic Cost

Freezing $\phi$ is often a pragmatic response to systematic constraints in scientific measurement, so it should not be dismissed as technical debt. Yet under indirect observation this pragmatism is not epistemically neutral: committing to a single $\phi$ freezes the evidential pipeline, which should be governed rather than treated as a default.

**Measurement heterogeneity creates pressure to standardize.** Instruments and acquisition protocols (e.g., scan-

ners, sensors, sites) induce systematic environment shifts that must be mapped into a comparable representation for learning (Thanjavur et al., 2021; Claverie et al., 2018; Yamashita et al., 2019). Standardization thus becomes the de facto interface enabling pooling, benchmarking, and reuse across environments (Gorgolewski et al., 2016; Markiewicz et al., 2021). As reuse scales, this interface turns $\phi$ into a shared community commitment rather than an individual analysis choice.

**Access and operational constraints enforce derivative-only interfaces.** Raw measurements are frequently inaccessible to downstream researchers due to consent/privacy restrictions, proprietary formats, or sheer scale, so communities distribute processed datasets as the practical interface for reuse (Mohr et al., 2025). Even when raw measurements are available, recomputing and benchmarking $P_\phi$ across plausible pipelines can be computationally and operationally prohibitive, effectively rendering pipeline choices irreversible for most users and making a frozen $\phi$ a structural consequence rather than a mere convention (Dugré et al., 2025; Allen et al., 2024).

## 2.3. Evidence Accumulation Without Lens Governance

These constraints motivate several practices for accumulating evidence and partially governing analysis paths. First, scientific communities rely on *cumulative evidence synthesis*, including systematic reviews (Mulrow, 1994), meta-analyses (Field & Gillett, 2010), and cross-study harmonization protocols (Arroyo-Araujo et al., 2022; Miller, 2017), to assess whether reported effects survive setting shifts and heterogeneous measurement conditions. Second, a growing methodology explicitly quantifies analysis-path sensitivity via multiverse analyses (Bell et al., 2022; Steegen et al., 2016), specification-curve style robustness checks (Simonsohn et al., 2020), and many-analysts exercises (Silberzahn et al., 2018), treating defensible analytic variation as an ob-

ject of inference rather than a nuisance to be hidden. Third, scientific communities increasingly rely on pipeline benchmarking to standardize practice and expose failure cases under controlled conditions (Brooks et al., 2024). This includes reference toolchains (Hirsch & Gilad-Bachrach, 2021; Klarner et al., 2023; Caucheteux et al., 2021), shared challenges (Zahedi et al., 2016), and comparative evaluations of preprocessing variants (Thiyagalingam et al., 2022).

However, these mechanisms impose different and incomplete forms of governance. Evidence synthesis is typically constrained to what prior studies have released or reported (Smith et al., 2016; Ioannidis, 2005), so it rarely operationalizes an admissible observation-model collection or propagates uncertainty over $\phi$ into new mechanistic claims. Multiverse and many-analysts designs do directly surface specification sensitivity, including sensitivity to pipeline choices, but they are typically deployed as study-level robustness audits and rarely yield standardized, community-wide constraints on admissible $\phi$ or reusable uncertainty summaries that downstream users can carry forward (Harder, 2020; Breznau et al., 2022). Pipeline benchmarking can compare alternative $\phi$ under controlled evaluations, yet it is commonly organized around proxy predictive or reconstruction objectives and does not, by default, require scientific validity criteria (e.g., invariance targets, transport conditions, or uncertainty accounting) to be made explicit (Maier-Hein et al., 2018; Roß et al., 2023). In this sense, current practice improves transparency around the interface without fully upgrading it into a governed observation model. For many downstream users, $\phi$ remains effectively fixed, and mechanistic conclusions can still inherit unscored dependence on upstream choices.

A related formal line of work asks how uncertain evidence should enter probabilistic inference. Jeffrey's probability-kinematics framework updates beliefs when evidence shifts probabilities over observable events rather than fixing a single observed event (Jeffrey, 1990). Pearl's virtual-evidence formulation instead represents soft evidence through likelihood-weighted auxiliary observations in a probabilistic graphical model (Pearl, 2014). More recent work on stochastic conditioning formalizes distributional evidence, where inference conditions on the event that an observable variable follows a distribution rather than on a realized value (Tolpin et al., 2021). Munk et al. compare these interpretations and show that treating the same uncertain evidence as a density over observations, a distributional event, or a likelihood-style soft observation can yield different posterior inferences (Munk et al., 2023).

This uncertain-evidence line of work provides the closest formal precedent for our argument. It shows that evidence is not only a value to be conditioned on, but an object whose uncertainty semantics must be specified before posterior inference is well defined. In indirect-observation workflows, this issue begins even earlier: the pipeline $\phi$ constructs the representation that later enters inference as evidence. Uncertainty over admissible $\phi$ is therefore observation-model uncertainty. If admissible $\phi$ is not represented explicitly, this uncertainty is collapsed before inference begins, leaving downstream users with a single released artifact rather than a recoverable evidential record.

## 3. From Frozen Lenses to Brittle Evidence

A frozen pipeline turns admissible observation-model variation into an artifact. Downstream optimization can improve performance on that artifact while leaving the mechanistic claim conditional on an untested pipeline commitment. This section explains how that commitment produces failure modes: **(C1)** hidden hypothesis space, **(C2)** uncertified transportability, and **(C3)** ungoverned multiplicity.

### 3.1. (C1): Hidden Hypothesis Space

**(C1)** concerns missing specification of the observation model. Downstream work often begins from a released representation without a recoverable account of the pipeline choices and validity conditions that produced it (Nichols et al., 2017). This omission is consequential: defensible pipeline choices can yield qualitatively different mechanistic conclusions (Kale et al., 2019). In AI4Science, such choices are not merely downstream "analysis variants"; they are alternative observation models that change how raw measurements are mapped into the released artifact (Murata et al., 2023; Steegen et al., 2016).

A common narrative attributes such dispersion primarily to human factors (p-hacking, inconsistent practice, or "researcher degrees of freedom") (Simmons et al., 2011). The dominant constraint is structural: the shared artifact is a single derivative, so admissible pipeline variation is not part of what can be jointly examined, reproduced, or compared across analysts. In ML terms, freezing the pipeline fixes the effective hypothesis space: the learner can only search over explanations compatible with the released representation (D'Amour et al., 2022), while the dependence of those explanations on admissible upstream choices is no longer recoverable from the released data alone (**Appendix B.2**).

The consequence is not that a pipeline-induced proxy is avoidable, but that mechanistic conclusions are often reported without qualifying the pipeline commitment that made them (Kiar et al., 2024). A model may fit a frozen derivative extremely well by exploiting pipeline-specific artifacts or inductive biases. Yet the released interface often does not make the pipeline or its validity conditions recoverable to downstream users, so predictive success on the artifact cannot be straightforwardly interpreted as regime-

stable mechanistic evidence (Carp, 2012).

## 3.2. (C2): Uncertified Transportability

**(C2)** arises when a documented pipeline lacks a tested regime of validity. Even when a community adopts a canonical pipeline, it rarely specifies when that pipeline is expected to fail as acquisition protocols, sensors, sites, or noise conditions change. Consequently, failures under distribution shift become hard to interpret: it is unclear whether they reflect a downstream learner's lack of robustness, a regime violation in the pipeline, or their interaction (D'Amour et al., 2022; Subbaswamy et al., 2021; Taori et al., 2020). This is therefore not only a generalization issue of the learner. It is a failure to certify the pipeline that defines what the downstream learners ever get to see.

A key reason that uncertified transportability is hard to "patch downstream" is *irreversible projection*. Many preprocessing operations are lossy and many-to-one (e.g., masking, coarse quantization, aggressive denoising), so distinct physical states can map to the same released artifact (Liu et al., 2020). Under regime shifts, even an identically specified pipeline can induce a different effective projection, collapsing distinctions that were discriminative in-regime and confounding mechanistic adjudication out-of-regime (Arridge et al., 2019; Stuart, 2010). If those distinctions are removed upstream, downstream learners cannot recover them from the released data (**Appendix B.3**). This enables proxy progress: learners can improve fit to a fixed surrogate while mechanistic conclusions stay brittle under admissible shifts (Subbaswamy et al., 2021; Maier-Hein et al., 2018).

Crucially, AI4Science tends to amplify this failure mode through repeated selection. As benchmarking and automated tuning scale, the workflow iterates adaptively against a fixed released derivative. Selection then rewards cues that predict the derivative well (including shortcuts), not cues that remain stable across regimes (Dwork et al., 2015; Blum & Hardt, 2015; Geirhos et al., 2020; Thiyagalingam et al., 2022). The result is a systematic transport gap: gains that are real on the released artifact may not accumulate as mechanistic knowledge across regimes.

## 3.3. (C3): Ungoverned Multiplicity

**(C3)** concerns the remaining multiplicity after specification and transport are addressed. Even if a pipeline is documented and restricted to a plausible validity regime, it is rarely unique: multiple defensible, regime-valid pipelines may remain. The failure is to report mechanistic conclusions as if this remaining variation had been resolved, rather than propagating it into uncertainty-qualified evidence and cross-study comparison.

This creates accumulation and optimization failures. Across studies, different admissible pipelines may imply different target claims, making synthesis ill-posed without a shared aggregation rule (Field & Gillett, 2010; Smith et al., 2016; Mulrow, 1994). Within a study, proxy metrics weakly constrain $\phi$: multiple admissible pipelines can tie on accuracy, AUC, or MSE yet yield different mechanisms, turning performance-based choice into ungoverned selection rather than uncertainty propagation (D'Amour et al., 2022; Bell et al., 2022). Appendix B.4 formalizes why defaulting or proxy-tuning a single pipeline behaves like criterion-misaligned selection over $\phi$.

Importantly, this is not a corner case. Many analyst and multiverse studies show that independent teams analyzing the same underlying data can reach meaningfully different quantitative conclusions under reasonable practice (Silberzahn et al., 2018; Botvinik-Nezer et al., 2020). This variability reflects genuine specification multiplicity, meaning that multiple defensible analysis choices can yield distinct results (Steegen et al., 2016; Breznau et al., 2022).

The constructive implication is that pipeline uncertainty should become an *input* to learning and reporting, not a hidden nuisance. Enumeration, while exposing sensitivity, fails to define a shared target or aggregation rule (Steegen et al., 2016; Simonsohn et al., 2020). End-to-end tuning, while optimizing proxy fit, implicitly collapses admissible variation into untracked selection (D'Amour et al., 2022; Bell et al., 2022; Dwork et al., 2015). Consequently, workflows should instead harness multiplicity as a *selection signal*, scoring pipelines by the stability of mechanistic conclusions across the admissible set (and across settings). If this uncertainty is not propagated, confidence is inflated and point claims replace sensitivity-qualified claims (Simmons et al., 2011; Ioannidis, 2005). The risk grows as AI4Science scales via shared derivatives, benchmarks, and automated selection on fixed interfaces (Thiyagalingam et al., 2022; Nichols et al., 2017). To quantify the severity of these blind spots, we turn from theoretical analysis to empirical auditing.

## 4. An Empirical Audit

We empirically operationalize the frozen-lens critique in EEG functional connectomics, a domain where neural activity is observed only through multi-stage measurement and preprocessing. The goal of the audit is not to establish a new depression biomarker or to rank pipelines by predictive performance. Instead, the audit asks whether conclusions drawn from a released representation remain stable when admissible observation-model choices are varied. For each dataset, we evaluate **864** preprocessing pipelines and **284** connectivity operators, yielding MDD–HC effect estimates under a uniform testing protocol. We then ask which configurations are both significant within a dataset and stable across datasets in effect direction and relative ordering.

*Table 2.* **Audit specification and outcome.** We audit a literature-grounded space of preprocessing pipelines and connectivity operators across two datasets. Only one pipeline–operator configuration remains both significant within datasets and stable across datasets.

| Element | Instantiation in Audit |
|---|---|
| Datasets | Two independent datasets: **MODMA** and **CISIR**. |
| Scientific target | MDD–HC difference in hemispheric FC asymmetry, a subject-level scalar comparing left–right symmetric connectivity structure. |
| Preprocessing pipelines | **864** literature-derived preprocessing configurations, each mapping raw EEG time series to a model-ready signal through choices such as filtering, referencing, denoising, downsampling, and windowing. |
| Operators | **284** SPI operators, each defining a rule for converting a pair of EEG channel time series into one FC edge weight. |
| Audited configurations | $864 \times 284 = \mathbf{245{,}376}$ pipeline–operator attempts. |
| Validation criteria | **Significance:** within-dataset MDD–HC separation under a fixed testing protocol. **Stability:** cross-dataset agreement in effect direction and relative ordering under bootstrap-supported assessment. |
| Outcome | Many configurations are significant in at least one dataset, but only one satisfies the joint criteria in both datasets $(1/245{,}376 \approx 0.0004\%)$. |

**Why this audit matters for (C1)–(C3).** This audit is not a performance benchmark. It is a stress test of whether a frozen data interface can support stable mechanistic evidence. By varying admissible preprocessing and connectivity choices, the sweep makes three quantities observable. First, if many defensible $\phi$ yield different significant effects, then the effective hypothesis space is pipeline-indexed and hidden behind the released representation (**C1**). Second, if effects that are significant in one dataset flip direction or reorder across an independent dataset, then the pipeline lacks certified transportability across measurement regimes (**C2**). Third, if many $\phi$ appear acceptable within a dataset but only a vanishing fraction remain stable across datasets, then multiplicity is real and should be propagated rather than collapsed to a default (**C3**).

### 4.1. Audit Protocol

*Table 2* summarizes the audit components. Here we state the minimal definitions needed to interpret the landscape, with full protocol details deferred to **Appendix C**.

**Terminology.** In this audit, functional connectivity (FC) denotes a matrix whose entries summarize statistical coupling between pairs of EEG channels (Zhan et al., 2026a; Chen et al., 2024b). Because both datasets are resting-state EEG recordings, we use resting-state functional connectivity

(rsFC) when emphasizing the acquisition condition and FC when referring to the matrix representation. Hemispheric asymmetry denotes a subject-level scalar summary obtained by comparing FC edge weights at left–right symmetric positions. The MDD–HC contrast is the group difference in this scalar summary between participants with major depressive disorder (MDD) and healthy controls (HCs), computed under a fixed estimator and testing protocol.

**Pipeline-induced representations.** For each dataset, we preprocess EEG time series with each pipeline to obtain a model-ready signal. Given this signal, we apply a downstream connectivity operator to construct a FC matrix (Luo et al., 2022). We use Statistical Pairwise Interactions (SPIs) as alternative rules for assigning an edge weight between two EEG channels. Concretely, an SPI takes two channel time series and returns one number for the FC edge. Pearson correlation is the familiar example (Cliff et al., 2023). Different SPIs therefore turn the same preprocessed EEG signal into different connectivity matrices, so each pipeline–SPI pair defines one admissible way of constructing the released representation. We score pipelines by (i) within-dataset significance using a Mann–Whitney $U$ test with multiple-comparison control (McKnight & Najab, 2010; Benjamini & Hochberg, 1995), and (ii) cross-dataset stability requiring bootstrap-supported directional agreement across datasets (DiCiccio & Efron, 1996).

**Significance and stability.** Significance refers to within-dataset evidence that a pipeline–SPI configuration separates MDD and HC participants under the fixed MDD–HC contrast. Stability refers to cross-dataset reproducibility of the same conclusion, requiring agreement in effect direction and relative ordering across MODMA and CISIR under bootstrap resampling.

### 4.2. Audit Result

*Figure 2* summarizes a slice of the audit: within-dataset significance versus bootstrap cross-dataset directional consistency of the MDD–HC hemispheric-asymmetry effect. A large fraction of configurations attain significance in at least one dataset, yet most fail to maintain consistent conclusions across datasets under the joint criteria. Across the full audited space of **245,376** pipeline–operator configurations, **only one** survives both significance and cross-dataset stability, yielding a survival rate of $\mathbf{1/245{,}376 \approx 0.0004\%}$ (conditional on the audited space and protocol).

### 4.3. Operationalizing (C1)–(C3) on the Landscape

Each pipeline–operator pair yields an MDD–HC effect estimate, turning (**C1**)–(**C3**) into an empirical landscape. Effects are pipeline-conditional because small admissible changes can reshuffle significance (**C1**). Many significant effects fail cross-dataset directional/rank stability, indicating

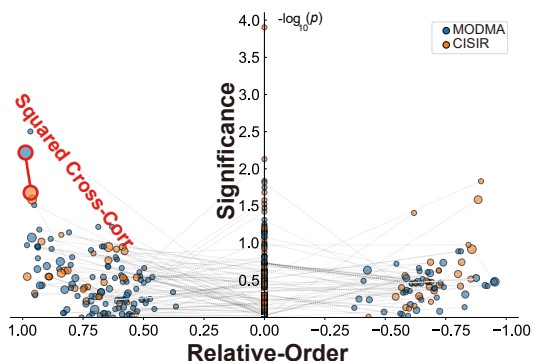

*Figure 2.* **Significance vs. cross-dataset stability.** Each point corresponds to an SPI evaluated in a dataset. Many SPIs appear significant in one dataset but fail joint criteria across datasets. Only one survives in both (Squared Cross-Corr, linked red circles).

regime-sensitive projection **(C2)**. Within-dataset winners are non-unique, so a single default hides multiplicity via untracked selection **(C3)**.

**Scope and Generalizability**. While our audit is implemented on EEG, **(C1)**–**(C3)** are structural under indirect observation. Recent benchmarking of brain connectivity-mapping methods further shows that downstream functional-connectivity organization depends substantially on the chosen mapping method (Liu et al., 2025; Luppi et al., 2024). While the $\approx 0.0004\%$ survival rate is specific to our audit, it exposes a fragility in AI4Science and should be read as a warning signal rather than an EEG-specific curiosity.

## 5. Alternative Views

Our position raises predictable concerns: If acknowledged best-practice toolchains already standardize pipelines, what is missing? If auditing requires large design-space searches, how can it be practical? If pipeline is mainly scientific craft, what is ML adding? We use these objections to sharpen the infrastructure question: what AI4Science lacks is not another checklist, but a way to make observation-model variation a computable object.

### 5.1. Objection A: "Best-practice toolchains already standardize the observation model"

A reasonable view is that indirect-observation fields already achieve comparability through widely adopted toolchains, quality control (QC) conventions, and benchmark-style reuse of shared derivatives (Esteban et al., 2019; Gorgolewski et al., 2016). On this view, pipelines are not truly "ungoverned": community norms and software defaults implicitly constrain choices, and remaining variation reflects analyst judgment rather than an epistemic gap.

**Why this view is incomplete.** Our concern is that standard-

ization freezes the pipeline upstream, outside the inference contract: downstream models condition on a released derivative as "given data" rather than treating the observation mapping as an inferential object. Defaults and QC provide procedural regularity, but they rarely define claim-aligned validity criteria (e.g., stability under admissible reprocessing and across settings) with an auditable sensitivity/stability summary. Consequently, multiple "best-practice" choices can yield different mechanistic conclusions, even when teams operate within the same shared interface (Botvinik-Nezer et al., 2020; Luppi et al., 2024).

**Our upgrade.** Keep the toolchains, but make the pipeline an ML object: define an explicit admissible pipeline space and score candidates by cross-setting claim stability (and sensitivity) under a declared compute budget. The output is not merely a default derivative, but a selected (or Pareto-efficient) pipeline/policy plus a stability profile that makes uncertainty over $\phi$ comparable across labs and time.

### 5.2. Objection B: "This is impractical at scale"

A pragmatic objection is that the combinatorial nature of search-based governance makes it computationally intractable and resource-intensive. Enumerating pipelines and operators, auditing stability across settings, and maintaining executable specifications can be prohibitive for typical lab budgets. On this view, the only workable compromise is to standardize the pipeline and focus optimization downstream.

**Why this view is incomplete.** The choice is not between exhaustive multiverse enumeration and a single frozen proxy (Steegen et al., 2016). What is impractical today is *bespoke* pipeline governance: uncertainty is absorbed by ad hoc analyst iteration and post hoc narrative selection rather than exposed as a computable object. Meanwhile, large compute is still spent on downstream winner-takes-all optimization against proxy metrics induced by a fixed pipeline. As a result, failures under reprocessing or setting shifts remain non-diagnosable (D'Amour et al., 2022). The scaling bottleneck is therefore not "too much computation," but the lack of a reusable representation that makes observation-model variation computable.

**Our upgrade.** We build on existing workflow and provenance infrastructure, which already makes realized pipelines executable, portable, and reproducible (Crusoe et al., 2022). For observation-model governance, the specification must also encode the admissible family of pipeline choices that could have produced the released representation. AI4Science should therefore standardize machine-readable pipeline specifications containing an executable workflow graph, typed operator parameters, admissible parameter domains, and provenance records for generated derivatives. With such specifications, automated systems can (i) search within $\Phi$ under explicit budgets, (ii) run

stability queries, and (iii) release portable audit records that others can reuse without re-implementing tool-specific pipelines (Feurer et al., 2015). This shifts "impractical at scale" into a platform objective: make pipelines executable, traceable, and queryable, so governance becomes amortized infrastructure rather than manual reporting.

### 5.3. Objection C: "This is not an ML problem"

A plausible critique is that pipeline governance is methodology or standards work, not an ML problem. The right solution is to publish better pipelines and reporting guidelines, rather than introducing new learning machinery.

**Why this view is incomplete.** Under indirect observation, the pipeline is a high-dimensional, interacting design space. In practice, the space is too large and too coupled for manual curation to yield a uniquely defensible interface. "Best practice" rarely specifies how to trade off admissibility, cross-setting transport, and sensitivity. This is precisely the regime where ML contributes: representing design spaces, defining stability-aligned objectives, and performing budgeted search and uncertainty-aware selection in a way that is reproducible and comparable across studies (D'Amour et al., 2022).

**Our upgrade.** ML contributes the missing inference machinery and aligned decision rules that can select or aggregate across admissible pipelines while propagating residual pipeline uncertainty into the reported claim. As AI4Science shifts toward reusable derivatives and automated experimentation, implicit pipeline choices become a dominant, scalable source of irreproducible mechanistic claims (Tom et al., 2024; Musslick et al., 2025). This makes governed, stability-first selection a first-class learning problem.

## 6. Call to Action

To make our position operational, we call on the AI4Science community to treat the pipeline as a queryable layer of the inference workflow. Concretely, we advocate shifting from procedural "best practice" checklists toward domain-specific Computable Observation Frameworks. This is not a proposal to replace existing toolchains or workflow standards. It is a proposal to expose their degrees of freedom as computable objects, so domains can evaluate, select, and report observation models with the same rigor that ML applies to predictive models.

A minimal implementation does not require fully automated end-to-end science. It requires three artifacts. First, a versioned specification of the admissible pipeline family $\Phi$, including operator choices, parameter domains, assumptions, and failure modes. Second, a bounded audit protocol that evaluates selected $\phi \in \Phi$ under declared datasets, settings, validity criteria, and compute budgets. Third, a reporting artifact that records which variation was explored, which

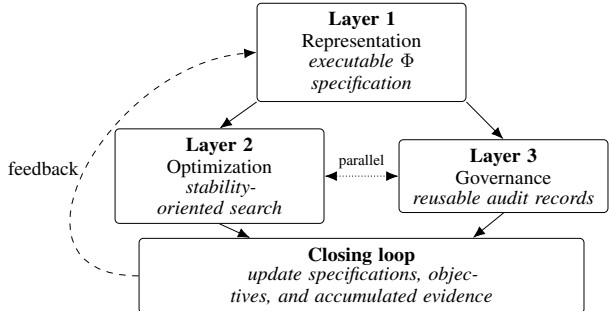

*Figure 3.* Flow of a Computable Observation Framework. **Layer 1** makes admissible variation executable as a specification of $\Phi$. **Layer 2** and **Layer 3** proceed in parallel, searching over $\Phi$ and accumulating audit records. The closing loop feeds audit outcomes back into the specification, objectives, and evidence base.

conclusions were stable, and which residual uncertainty over $\phi$ remains. *Figure* 3 summarizes this workflow and the dependency among layers. *Table* 3 specifies who should do what across domain science, ML methodology, and research infrastructure.

### 6.1. Layer 1: Representation

**Layer 1** is a **versioned, executable pipeline specification**: a machine-readable operator graph with typed nodes, constrained parameter domains, software implementations, provenance records, and explicit semantics about assumptions and failure modes. This specification turns hidden configuration choices into explicit objects that can be executed, inspected, replayed, and varied under declared admissible choices. Crucially, it remains compatible with existing toolchains, serving as an export target for workflow DAGs rather than a replacement for them.

**Atomic observation operators.** The specification should expose a typed operator interface with declared assumptions and failure modes, starting from a small, high-impact subset of operators. Existing toolchains can be integrated via thin wrappers that export these operators and their parameters into the specification, enabling comparison and controlled counterfactual re-execution without full re-implementation.

**Computable pipeline specifications.** The relevant requirement is that pipeline variation can be executed and queried systematically. A practical specification can be a typed workflow DAG: each node names an operator, points to its software implementation, declares input–output types, and lists admissible parameter domains. For example, an EEG node can expose referencing as a categorical choice (average versus REST), ICA removal as a binary choice, and band-pass cutoffs as discrete or continuous parameters. This representation supports differentiable tuning where available, but also gradient-free search, rule-based enumeration, and policy-based selection over discrete, conditional, or

*Table 3.* Division of expertise for implementing Computable Observation Frameworks.

| Expertise | Main role | COF artifact |
|---|---|---|
| Domain scientists | Specify admissible Φ: defensible operators, parameter ranges, validity conditions, and failure modes. | Domain-constrained Φ specification. |
| ML researchers | Make Φ computable: DAG/spec exports, mixed search, stability objectives, and uncertainty-aware selection. | Search and selection procedures over Φ. |
| Methodologists / infrastructure | Make audits reusable: provenance, audit interfaces, reporting templates, and cumulative evidence bases. | Reusable audit records and reports. |

tool-specific choices.

### 6.2. Layer 2: Optimization

Once pipelines are representable, the relevant computation goal is to evaluate and choose pipelines by claim validity: optimizing for conclusions that remain stable under admissible reprocessing and transportable across settings.

**Stability objectives.** Transport and insensitivity to nuisance variation must be formulated as explicit computational targets, with low failure rates under declared perturbations. The optimization loop then searches for configurations $\phi$ that maximize these criteria across admissible variation and settings, rather than merely maximizing predictive fit within a single frozen artifact.

**Acceleration and amortization.** Practical stability search requires amortization primitives: reusing compiled graphs, exploiting structure-aware subsampling of benchmarks (Zhan et al., 2026b), and caching intermediate artifacts so audits become progressively cheaper across studies. The deliverable is a Pareto set (or a policy over Φ) equipped with a stability/sensitivity profile under a declared compute budget, rather than a single unqualified point commitment.

### 6.3. Layer 3: Governance

**Layer 3** is a governance platform that provides shared computation and shared memory for the domain. The platform runs audits reproducibly and accumulates their outputs into a reusable evidence base.

**Audited execution at scale.** Given a pipeline specification (**Layer 1**) and a stability objective (**Layer 2**), the platform runs declared audit protocols across datasets and settings, logging intermediate artifacts and preserving provenance traces. This converts what is currently bespoke analyst iteration into a standardized, verifiable scientific computation.

**Evidence accumulation over observation-model space.** The platform accumulates audit outputs in a queryable form that supports claim evaluation: given a target claim and declared admissibility, it estimates the claim's confidence under admissible pipeline variation and records which regions of Φ support or refute the claim. In this way, later

studies can reuse and update accumulated evidence rather than re-discovering fragile degrees of freedom via bespoke iteration.

**Cross-study comparability.** Because pipelines are expensive to rerun and hard to compare across heterogeneous toolchains, operating on a shared pipeline specification enables counterfactual queries (e.g., varying $\phi$ within a declared admissible neighborhood) without bespoke re-implementation. This makes disagreements diagnosable: the community can test whether conflicting conclusions arise from measurement variation, dataset shift, or observation-model incompatibility induced by lossy upstream choices (Crusoe et al., 2022).

### 6.4. Closing the Loop

Adopting the Computable Observation Framework makes scientific evidence accumulative, addressing **(C1)**–**(C3)** in one workflow: **Layer 1** turns pipelines into typed, versioned, executable objects, **Layer 2** makes cross-setting transport a first-class optimization target via stability-first objectives, and **Layer 3** reproduces and aggregates audits across datasets to localize failure modes and accumulate stability profiles, so unavoidable multiplicity is propagated as uncertainty (Pareto sets or policies over Φ) rather than collapsed to an untracked point choice.

## 7. Conclusion

**We take a simple position: AI4Science should treat measurement-to-dataset pipelines as inference components.** This is a continuation of recent position works on Data-centric ML (Villalobos et al., 2024; Kandpal & Raffel, 2025) and AI4Science (Hogg & Villar, 2024). Importantly, we do not advocate any single inferential doctrine (e.g., fully Bayesian (Stuart, 2010) or multiverse analysis (Steegen et al., 2016)), nor a simple "pipeline AutoML" (He et al., 2021). Instead, we call for building a Computable Observation Framework that turns observation-model uncertainty into *cumulative evidence*. This is more pressing as foundation models, agents, and automated laboratories accelerate scientific iteration (Romera-Paredes et al., 2024; Tom et al., 2024; Chenthamarakshan et al., 2023).

## Acknowledgements

This work is supported by Natural Science Foundation of China (No. 72374173), the Fundamental Research Funds for the Central Universities (No. SWU-XDJH202303), University Innovation Research Group of Chongqing (No. CXQT21005), China Scholarship Council (CSC) program (No. 202306990091, No. 202406990056) and Chongqing Graduate Research Innovation Project (No. CYB22129, No. CYB240088). The experiments are supported by the High Performance Computing clusters and the Large-Scale Instrument Sharing Platform (H20 GPU Server) at Southwest University. We acknowledge the opensource data providers and the `pyspi` developer team for their important contributions to the open scientific community. We are also grateful to the reviewers for their constructive and insightful comments, which helped improve the quality and presentation of this work.

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

# A. Pipeline Exemplars Across Indirect-Observation Sciences

*Figure 4* summarizes five high-impact domains where the community commonly releases a *processed derivative* as the practical interface for reuse, benchmarking, and downstream inference. In each case, the derivative is produced by a multi-stage preprocessing/reconstruction pipeline whose parameterization encodes consequential, convention-driven modeling choices. Freezing a single default pipeline therefore operationalizes a *frozen lens*: it fixes an observational operator upstream, while leaving the induced analysis-path uncertainty and setting dependence under-governed.

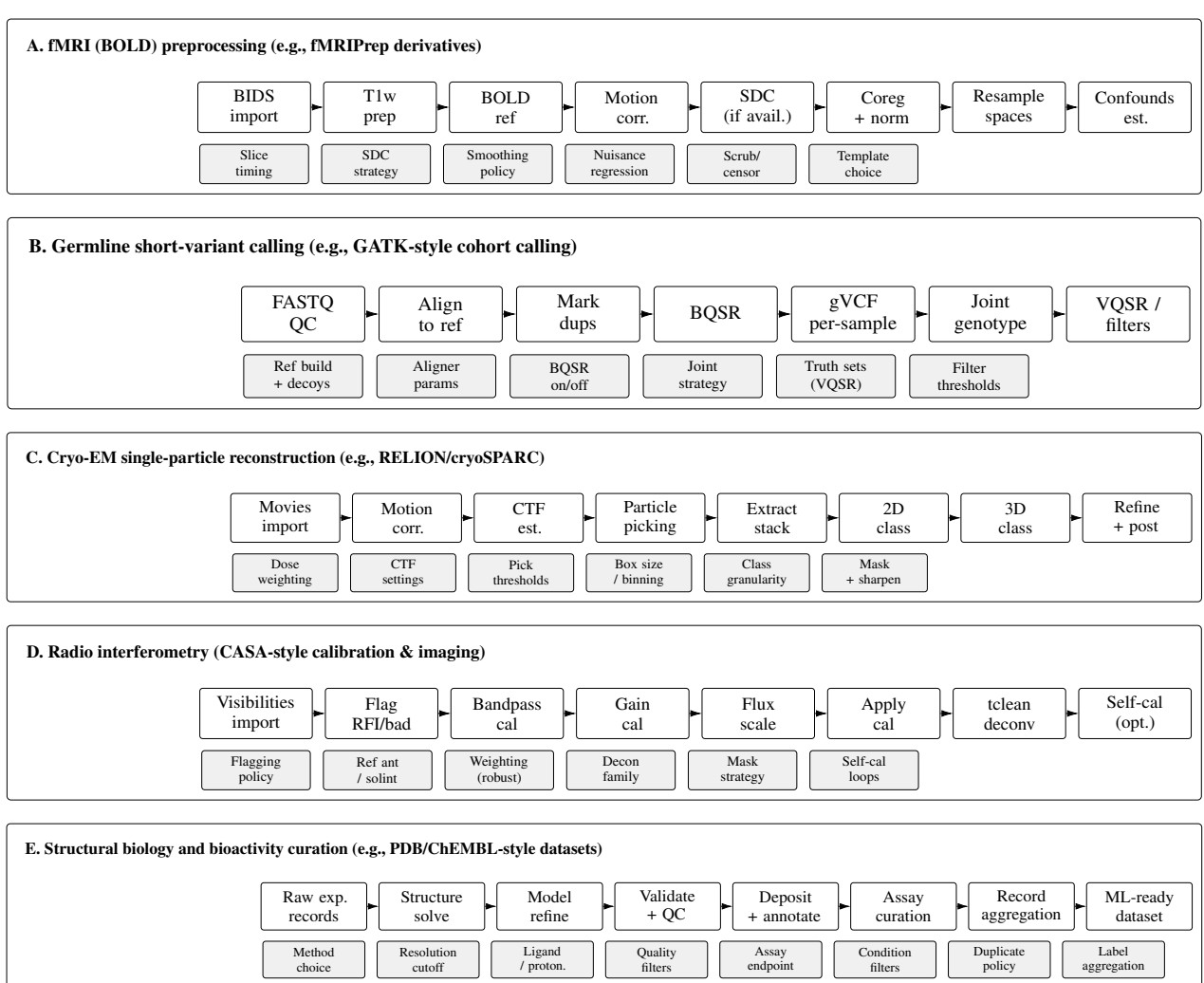

*Figure 4.* Canonical preprocessing/reconstruction pipelines in five indirect-observation domains. Top row in each panel: typical operational steps from raw measurements or primary experimental records to the released derivative artifact. Bottom row: representative fork points that are commonly fixed by defaults, conventions, or tool choices, thereby "freezing" an observation model and inducing analysis-path uncertainty.

**fMRI (BOLD).** Neuroimaging datasets are often distributed in standardized layouts (BIDS) and processed into derivatives using automated pipelines such as fMRIPrep (Gorgolewski et al., 2016; Esteban et al., 2019). Canonical steps include anatomical preprocessing and spatial normalization, motion and distortion correction, resampling to standard spaces, and confound estimation (Esteban et al., 2019). Crucially, downstream conclusions can vary substantially across defensible analysis choices ("many analysts") even with the same raw dataset (Botvinik-Nezer et al., 2020). This exactly mirrors our EEG findings and motivates explicit accounting of pipeline degrees of freedom (e.g., SDC strategy, smoothing, nuisance regression, censoring).

**Germline short-variant calling (WGS/WES).** A widely used cohort pipeline follows: read QC and alignment, duplicate marking and (optionally) base quality score recalibration, per-sample gVCF calling, joint genotyping, and callset filtering via VQSR (or hard filters) (McKenna et al., 2010; DePristo et al., 2011; Van der Auwera et al., 2013). Pipeline choices (reference build, aligner/caller stack, BQSR usage, and filtering regime) can shift error profiles and thus propagate into association and interpretation (Zhao et al., 2020; Betschart et al., 2022; Olson et al., 2022). Hard filtering thresholds here act as uncertified projections, creating the same **(C2)** Uncertified Transportability risk we measure in **Section 3**.

**Cryo-EM single-particle reconstruction.** Single-particle cryo-EM pipelines proceed from dose-fractionated movies through motion correction and CTF estimation to particle picking/extraction, 2D/3D classification, refinement, and post-processing (including masking/sharpening and validation) (Scheres, 2019; Fernandez-Leiro & Scheres, 2017; Punjani et al., 2017). These steps constitute an explicit inference pipeline, with well-known sensitivity to choices such as picking thresholds, class granularity, and post-processing settings.

**Radio interferometric astronomy (visibility → image).** Radio data reduction typically calibrates raw visibilities via flagging, bandpass and gain calibration, flux scaling, application of solutions, and imaging/deconvolution (e.g., CLEAN-family reconstruction in `tclean`), often with optional self-calibration loops (Bean et al., 2022; Hunter et al., 2023). Key pipeline decisions include flagging policy, calibration solution intervals, imaging weighting (e.g., Briggs robust), and masking/deconvolver choices, which reshape the effective measurement operator and recovered morphology.

**Structural biology and bioactivity databases.** The frozen-lens dynamic also appears when downstream AI4Science models are trained on curated scientific databases rather than on a single laboratory pipeline. Protein Data Bank (PDB) entries are not raw molecular reality: they are deposited structural models produced through experimental acquisition, reconstruction, refinement, validation, and curation choices (Berman et al., 2000; wwPDB Consortium, 2019). Likewise, ChEMBL-style bioactivity records aggregate measurements across assays whose targets, protocols, endpoints, concentrations, cell lines, organisms, and normalization conventions may differ substantially (Gaulton et al., 2017; Mendez et al., 2019). For downstream drug-discovery models, these records often function as fixed training labels or structural inputs, but the released value is conditional on upstream experimental and curation choices. Thus, even when the database is community-standard, the interface may still collapse admissible observation-model variation: alternative assay filters, endpoint harmonization rules, structure-quality thresholds, or aggregation policies can change which molecular relationships are learnable from the released representation.

## B. Formalizing the Structural Failure Modes

This section formalizes why bundling the observation model $P_\phi$ into the dataset definition induces structural failure modes under indirect observation. We focus on the common AI4Science regime where $P_\phi$ is fixed upstream, and use minimal notation to sharpen three claims: **(C1)** hidden hypothesis spaces induced by the pipeline, **(C2)** uncertified transportability via irreversible evidence projection, and **(C3)** ungoverned multiplicity induced by pipeline dispersion and estimand mismatch.

### B.1. Setup: artifacts are pipeline-indexed, and accumulation requires a shared estimand

Let $x \in \mathcal{X}$ denote raw measurements, $\psi \in \Psi$ the scientific quantity of interest, and $s \in \mathcal{S}$ index settings (e.g., measuring sites, cohorts, protocols, instruments). Under indirect observation, the released representation is produced by a pipeline (observation model)

$$z_\phi = P_\phi(x), \qquad \phi \in \Phi, \tag{1}$$

where $\Phi$ is the scientifically *admissible* set of pipelines. A released dataset is a single realized artifact $z_{\mathrm{rel}} = z_{\phi_0}$ for some (often implicit) committed $\phi_0$. As a result, conditioning on a particular released artifact $z_{\mathrm{rel}}$ implicitly conditions on the pipeline that produced it.

Let $y$ denote the proxy task target or observed task-level outcome. Given an artifact $z_\phi$, a downstream model produces

$$\hat{y} = F_\theta(z_\phi),$$

and is typically optimized within a setting $s$ under a proxy loss, where $D_s$ denotes the setting-$s$ distribution over $(x, y)$:

$$\theta^*(\phi, s) = \arg\min_\theta \mathbb{E}_{(x,y) \sim D_s} \left[ L(F_\theta(P_\phi(x)), y) \right]. \tag{2}$$

**Key distinction.** Standard workflows treat $\phi$ as fixed upstream and optimize only $\theta$. Our position is that $\phi$ is a scientific commitment that determines what evidence is present in the artifact, so comparability and accumulation require explicit treatment of pipeline dispersion rather than implicit point commitments.

## B.2. C1: Hidden Hypothesis Spaces

Pipeline specifies a representation map $P_\phi : \mathcal{X} \to \mathcal{Z}$. Let $\mathcal{F} = \{ z \mapsto F_\theta(z) : \theta \in \Theta \}$ denote the downstream model family. For any fixed $\phi$, the induced (measurement-to-science) hypothesis class is

$$\mathcal{H}_\phi = \left\{ h : \mathcal{X} \to \Psi \;\middle|\; h(x) = f(P_\phi(x)) \text{ for some } f \in \mathcal{F} \right\}. \tag{3}$$

Thus choosing $\phi$ is choosing $\mathcal{H}_\phi$. Unlike soft regularization, many preprocessing operators are non-invertible (e.g., filtering, projection), so they impose *hard* constraints: if the true mechanism $h^* \notin \mathcal{H}_\phi$, the misspecification is structural.

Over admissible pipelines, the effective hypothesis space is the union

$$\mathcal{H}_\Phi = \bigcup_{\phi \in \Phi} \mathcal{H}_\phi, \tag{4}$$

which bundling collapses to a single implicit $\mathcal{H}_{\phi_0}$.

**Takeaway (C1).** Pipelines define hidden hypothesis spaces: bundling fixes an implicit $\mathcal{H}_{\phi_0}$ and imports strong compositional inductive biases while obscuring the effective search space $\mathcal{H}_\Phi$ in (4).

## B.3. C2: Uncertified transportability (via irreversible evidence projection)

A pipeline can irreversibly collapse measurements onto a representation manifold implied by its assumptions *before* modeling. Formally, for a given $\phi$, distinct measurement states $x_1 \neq x_2$ can be mapped to indistinguishable representations

$$P_\phi(x_1) \approx P_\phi(x_2). \tag{5}$$

Equivalently, $P_\phi$ induces an equivalence relation on measurements,

$$x_1 \sim_\phi x_2 \iff P_\phi(x_1) = P_\phi(x_2). \tag{6}$$

Fixing $P_\phi$ enforces that all variation within an equivalence class is treated as nuisance at the representation level. If the scientific target is not invariant under $\sim_\phi$ (i.e., $\exists\, x_1 \sim_\phi x_2$ with $\psi(x_1) \neq \psi(x_2)$), then the mechanistically relevant distinction is irrecoverable from $z_\phi$ by any downstream model that only observes $P_\phi(x)$.

**Takeaway (C2).** Freezing a pipeline commits measurements to a quotient space (6) *before* learning, so predictive gains can certify the pipeline rather than the mechanism, potentially discarding evidence for competing explanations upstream.

## B.4. C3: Ungoverned multiplicity (pipeline dispersion and estimand mismatch)

Under indirect observation, $\Phi$ often contains multiple scientifically defensible pipelines. Within a setting $s$, many $\phi$ can yield similar proxy performance after downstream optimization:

$$R_s(\phi) = \min_\theta \mathbb{E}_{(x,y) \sim D_s} \left[ L(F_\theta(P_\phi(x)), y) \right], \qquad R_s(\phi_1) \approx R_s(\phi_2). \tag{7}$$

Yet mechanistic conclusions can differ across pipelines even when $\mathcal{R}_s(\phi)$ is comparable. Let $C(\phi, s)$ denote a mechanistic output induced by the full pipeline (e.g., feature attribution, connectivity pattern, inferred causal edge set). Then **pipeline dispersion is consequential** when there exist $\phi_1, \phi_2 \in \Phi$ such that

$$\mathcal{R}_s(\phi_1) \approx \mathcal{R}_s(\phi_2) \quad \text{but} \quad C(\phi_1, s) \neq C(\phi_2, s). \tag{8}$$

In this regime, $\phi$ is weakly identified by the within-setting proxy objective: proxy fit does not determine the mechanistic conclusion.

**Estimand mismatch and the crisis of accumulation.** A released artifact $z_{\phi_0}$ induces a pipeline-indexed object of inference (implicitly $C(\phi_0, s)$). Across studies that use different admissible pipeline configurations $\phi$ (including different defaults,

toolchains, or "equivalent" pipelines), the community is not estimating a single shared quantity but a *family* $\{C(\phi, s)\}_{\phi \in \Phi}$. Thus cross-study agreement and meta-analytic aggregation are well-defined only if the community supplies an explicit rule that either: (i) standardizes $\phi$ (fixing the estimand), or (ii) propagates pipeline uncertainty by combining evidence over $\Phi$ (fixing the estimand by marginalization or an agreed summary functional). This missing rule is a governance gap that leaves accumulation ill-posed rather than merely noisy. Formally, if one aims to interpret conclusions as evidence about the scientific target $\psi$ from raw measurements $x$, a generic uncertainty-propagating form is

$$P(\psi \mid x) \ \propto \ \int_{\phi \in \Phi} P(\psi \mid z_\phi) \, P(\phi) \, d\phi, \qquad z_\phi = P_\phi(x). \tag{9}$$

Without such a standardization or aggregation rule, different studies need not share a common estimand, so "replication" becomes ill-posed rather than merely noisy.

**Selection is not propagation.** Current practice resolves pipeline dispersion by selection (implicit defaults, ad-hoc tuning, or significance-based choice), yielding a point commitment $\hat{\phi}$ that may vary across settings and analysts. Selection can therefore convert admissible pipeline variation into hidden degrees of freedom while leaving mechanistic claims unqualified by pipeline sensitivity.

**Takeaway (C3).** Indirect observation induces a nontrivial admissible set $\Phi$; when (8) holds, proxy fit weakly identifies $\phi$ and does not determine mechanism. Absent a community rule that standardizes or propagates over $\Phi$ (e.g., (9)), this governance gap ensures studies do not share a common estimand, and accumulation is not well-defined.

## C. Motivating Empirical Audit

In this section, we present an empirical study that examines EEG-based hemispheric asymmetry in resting-state functional connectivity (rsFC) for Major Depressive Disorder (MDD). This question has been debated for decades (Van Der Vinne et al., 2017). Frontal alpha-band asymmetry has long been proposed as a candidate biomarker for MDD (Mumtaz et al., 2015). However, recent meta-analyses and multiverse analyses suggest that the effect is weak or not reliably observed (Kołodziej et al., 2021). We designed our study to revisit the question from a network perspective. In doing so, we find that plausible pipeline choices materially shape the inferred rsFC asymmetry, consistent with our broader claim that pipeline injects consequential inductive biases into the evidence used for downstream inference. We next describe the experimental setup and summarize the main findings.

### C.1. Dataset

**CISIR (Mumtaz, 2016)** This dataset was released by Centre for Intelligent Signal and Imaging Research, Universiti Teknologi PETRONAS, Malaysia in 2016. It includes 34 MDD outpatients (18 females) (age, mean=40.33, std±12.861) and an age-matched group of 30 healthy controls (HCs, 9 females) (age, mean=38.227, std±15.64). The MDD participants met the diagnostic criteria for MDD according to Diagnostic and Statistical Manual-IV (DSM-IV) (Do, 2011). They are recruited from the outpatient clinic of hospital Universiti Sains Malaysia (HUMS), Malaysia. The experiment design is approved by the ethics committee, HUSM. The MDD participants with psychotic symptoms, pregnant patients, alcoholics, smokers and patients having epileptic problems are excluded. The normal controls were screened for possible mental or physical illness and found to be free of neurological and psychiatric disease. All participants were instructed to abstain from caffeine, nicotine, alcohol before EEG recordings, and completed EEG acquisition at the same time of day.

**MODMA (Cai et al., 2022)** This dataset is released by Gansu Provincial Key Laboratory of Wearable Computing, School of Information Science and Engineering, Lanzhou University, China. This dataset includes 53 participants with 24 outpatients (females=11) (age, mean=30.09, std±10.368) and 29 HCs (females=9) (age, mean=31.4, std±8.99). All participants had a normal or corrected-to-normal vision. Patients with MDD were recruited from inpatient and outpatient services from Lanzhou University Second Hospital, Gansu, China, diagnosed by at least one attending psychiatrist. All MDD patients received a structured Mini-International Neuropsychiatric Interview (MINI) (Sheehan et al., 1998) that met the diagnostic criteria for major depression of DSM-IV. The inclusion criteria for all participants were the age between 18 and 55 years old and primary or higher education level. The HCs are recruited by posters. The study was approved by the Ethics Committee of the Second Affiliated Hospital of Lanzhou University, and written informed consent was obtained from all subjects before the experiment began. For MDD patients, inclusion criteria required meeting the diagnostic criteria for depression according to the MINI, participants' Patient Health Questionnaire-9 item (PHQ-9) (Spitzer et al., 1999) score

was greater than or equal to 5, and no psychotropic drug treatment within the prior two weeks. For MDD patients, the exclusion criteria were having mental disorders or brain organ damage, having a serious physical illness, and severe suicidal tendencies. For HC, the exclusion criteria included a personal or family history of mental disorders. Exclusion criteria for all subjects included abuse of or dependence on alcohol or psychotropic drugs in the past year, women who were pregnant or lactating, or taking birth control pills.

### C.2. Preprocessing Steps

**Common Settings**  We use MATLAB [1] with EEGLAB toolbox (Delorme & Makeig, 2004) for offline EEG preprocessing. We selected 19 common channels and applied the same preprocessing to the CISIR and MODMA datasets for consistency. We use a series of band-pass filters on the preprocessed EEG records. These frequency bands included delta (1-4 Hz), theta (4-8 Hz), alpha (8-13 Hz), beta (13-30 Hz), gamma (30-40 Hz), and the full band (1-40 Hz) according to previous studies (Cohen, 2014; Chen et al., 2024a).

**Preprocessing Choices**  The original papers recommend different preprocessing pipelines, so we audited the inductive bias introduced by admissible preprocessing variation. We evaluated 864 valid preprocessing configurations under a top-level factorial design: $2 \times 4 \times 3 \times 2 \times 3 \times 6 = 864$. The factors were:

- PREP (Bigdely-Shamlo et al., 2015): None, Automatic.

- Downsampling: Original, 250Hz, 200Hz, 100Hz.

- Filtering: None, [0.1Hz, 40Hz], [1Hz, 40Hz].

- Re-referencing: Average, REST (Dong et al., 2017).

- Artifact handling: None, automatic bad-channel rejection, or ICA+MARA (Haresign et al., 2021)+TrimOutlier.

- Window size: 100ms, 200ms, 400ms, 1s, 2s, 4s.

Bad-channel interpolation was applied only within the automatic bad-channel rejection branch and was not an independent design factor. For each valid preprocessing configuration, we ran the full set of 284 SPIs (Cliff et al., 2023), yielding $245,376$ candidate networks.

*Table* 4 lists the estimator-level SPIs used in the audit. The identifiers follow the version-locked `pyspi` configuration used in our experiments.

### C.3. Method

**Hemispheric Functional Connectomic Asymmetry**  Formally, the adjacency matrix $\mathbf{A}$ could be rewritten as:

$$\mathbf{A} = \begin{bmatrix} \mathbf{A}_{LL} & \mathbf{A}_{LR} \\ \mathbf{A}_{RL} & \mathbf{A}_{RR} \end{bmatrix}, \tag{10}$$

where $\mathbf{A}_{LL}$ and $\mathbf{A}_{RR}$ denote rsFC within left hemisphere and right hemisphere, respectively. $\mathbf{A}_{LR}$ denotes rsFC from left hemisphere to right hemisphere and $\mathbf{A}_{RL}$ denotes rsFC from right hemisphere to left hemisphere. For undirected SPIs, $A_{LR}$ and $A_{RL}$ are transposes under matched channel ordering. For directed SPIs, the two blocks can differ. We estimate the rsFC hemispheric asymmetry by calculating the differences in edge weights at symmetric positions between the left and right hemispheres and taking the median across edges. We use the median for robustness to outliers. Empirically, it also yields more consistent cross-dataset behavior than mean-based summaries.

**Significance and Consistency**  A reasonable validity criterion is that an inferred effect should be (i) *significant* within each dataset and (ii) *consistent* across independent datasets. Accordingly, we use two complementary indices to evaluate each pipeline–SPI configuration.

---

[1]https://www.mathworks.com

*Table 4.* Estimator-level Statistical Pairwise Interactions (SPIs) used in the EEG audit. Each identifier denotes one `pyspi` statistic that maps a pair of EEG channel time series to one scalar FC edge weight. The list follows the version-locked `pyspi` configuration used in the audit and contains the 284 SPIs evaluated in our experiments.

| | | | |
|---|---|---|---|
| cov_EmpiricalCovariance | cov_EllipticEnvelope | cov_MinCovDet | cov_GraphicalLasso |
| cov_GraphicalLassoCV | cov_ShrunkCovariance | cov_LedoitWolf | cov_OAS |
| prec_EmpiricalCovariance | prec_EllipticEnvelope | prec_MinCovDet | prec_GraphicalLasso |
| prec_GraphicalLassoCV | prec_LedoitWolf | prec_OAS | prec_ShrunkCovariance |
| spearmanr | spearmanr-sq | kendalltau | kendalltau-sq |
| xcorr-max_sig-True | xcorr_mean_sig-True | xcorr_mean_sig-False | xcorr-sq_max_sig-True |
| xcorr-sq_mean_sig-True | xcorr-sq_mean_sig-False | cov-sq_EmpiricalCovariance | cov-sq_EllipticEnvelope |
| cov-sq_GraphicalLasso | cov-sq_GraphicalLassoCV | cov-sq_LedoitWolf | cov-sq_MinCovDet |
| cov-sq_OAS | cov-sq_ShrunkCovariance | prec-sq_EmpiricalCovariance | prec-sq_EllipticEnvelope |
| prec-sq_GraphicalLasso | prec-sq_GraphicalLassoCV | prec-sq_LedoitWolf | prec-sq_MinCovDet |
| prec-sq_OAS | prec-sq_ShrunkCovariance | pdist_euclidean | pdist_cityblock |
| pdist_cosine | pdist_chebyshev | pdist_canberra | pdist_braycurtis |
| dcorr | dcorr_biased | dcorrx_maxlag-1 | dcorrx_maxlag-10 |
| mgc | mgcx_maxlag-1 | mgcx_maxlag-10 | hsic |
| hsic_biased | hhg | dtw | dtw_constraint-itakura |
| dtw_constraint-sakoe_chiba | lcss | lcss_constraint-itakura | lcss_constraint-sakoe_chiba |
| softdtw | softdtw_constraint-itakura | softdtw_constraint-sakoe_chiba | bary_euclidean_mean |
| bary_euclidean_max | bary_dtw_mean | bary_dtw_max | bary_softdtw_mean |
| bary_softdtw_max | bary_sgddtw_mean | bary_sgddtw_max | bary_sq_euclidean_mean |
| bary_sq_euclidean_max | bary_sq_dtw_mean | bary_sq_dtw_max | bary_sq_sgddtw_mean |
| bary_sq_sgddtw_max | bary_sq_softdtw_mean | bary_sq_softdtw_max | gwtau |
| anm | igci | cds | reci |
| ccm_E-1_mean | ccm_E-1_max | ccm_E-1_diff | ccm_E-10_mean |
| ccm_E-10_max | ccm_E-10-diff | ccm_E-None_mean | ccm_E-None_max |
| ccm_E-None_diff | je_gaussian | je_kozachenko | je_kernel_W-0.5 |
| ce_gaussian | ce_kozachenko | ce_kernel_W-0.5 | mi_gaussian |
| mi_kraskov_NN-4 | mi_kraskov_NN-4_DCE | mi_kernel_W-0.25 | tlmi_gaussian |
| tlmi_kraskov_NN-4 | tlmi_kraskov_NN-4_DCE | tlmi_kernel_W-0.25 | te_kraskov_NN-4_k-max-10_tau-max-4 |
| te_kraskov_NN-4_DCE_k-max-10_tau-max-4 | te_kraskov_NN-4_DCE_k-1_kt-1_l-1_lt-1 | te_kraskov_NN-4_DCE_k-2_kt-1_l-1_lt-1 | te_kraskov_NN-4_k-1_kt-1_l-1_lt-1 |
| te_kernel_W-0.25_k-1 | te_symbolic_k-1_kt-1_l-1_lt-1 | te_symbolic_k-10_kt-1_l-1_lt-1 | gc_gaussian_k-max-10_tau-max-2 |
| gc_gaussian_k-1_kt-1_l-1_lt-1 | cce_gaussian | cce_kozachenko | cce_kernel_W-0.5 |
| di_gaussian | di_kozachenko | di_kernel_W-0.5 | si_gaussian_k-1 |
| si_kozachenko_k-1 | si_kernel_W-0.5_k-1 | phi_star_t-1_norm-0 | phi_star_t-1_norm-1 |
| phi_Geo_t-1_norm-0 | phi_Geo_t-1_norm-1 | xme_kozachenko_k1 | xme_kozachenko_k10 |
| xme_gaussian_k1 | xme_gaussian_k1 | xme_kernel_k1 | xme_kernel_k10 |
| cohmag_multitaper_mean_fs-1_fmin-0_fmax-0-5 | cohmag_multitaper_mean_fs-1_fmin-0_fmax-0-25 | cohmag_multitaper_mean_fs-1_fmin-0-25_fmax-0-5 | cohmag_multitaper_max_fs-1_fmin-0_fmax-0-5 |
| cohmag_multitaper_max_fs-1_fmin-0_fmax-0-25 | cohmag_multitaper_max_fs-1_fmin-0-25_fmax-0-5 | phase_multitaper_mean_fs-1_fmin-0_fmax-0-5 | phase_multitaper_mean_fs-1_fmin-0_fmax-0-25 |
| phase_multitaper_mean_fs-1_fmin-0-25_fmax-0-5 | phase_multitaper_max_fs-1_fmin-0_fmax-0-5 | phase_multitaper_max_fs-1_fmin-0_fmax-0-25 | phase_multitaper_max_fs-1_fmin-0-25_fmax-0-5 |
| gd_multitaper_delay_fs-1_fmin-0_fmax-0-5 | gd_multitaper_delay_fs-1_fmin-0_fmax-0-25 | gd_multitaper_delay_fs-1_fmin-0-25_fmax-0-5 | psi_multitaper_mean_fs-1_fmin-0_fmax-0-5 |
| psi_multitaper_mean_fs-1_fmin-0_fmax-0-25 | psi_multitaper_mean_fs-1_fmin-0-25_fmax-0-5 | psi_wavelet_mean_fs-1_fmin-0_fmax-0-5_mean | psi_wavelet_mean_fs-1_fmin-0_fmax-0-25_mean |
| psi_wavelet_mean_fs-1_fmin-0-25_fmax-0-5_mean | psi_wavelet_max_fs-1_fmin-0_fmax-0-5_max | psi_wavelet_max_fs-1_fmin-0_fmax-0-25_max | psi_wavelet_max_fs-1_fmin-0-25_fmax-0-5_max |
| icoh_multitaper_mean_fs-1_fmin-0_fmax-0-5 | icoh_multitaper_mean_fs-1_fmin-0_fmax-0-25 | icoh_multitaper_mean_fs-1_fmin-0-25_fmax-0-5 | icoh_multitaper_max_fs-1_fmin-0_fmax-0-5 |
| icoh_multitaper_max_fs-1_fmin-0_fmax-0-25 | icoh_multitaper_max_fs-1_fmin-0-25_fmax-0-5 | plv_multitaper_mean_fs-1_fmin-0_fmax-0-5 | plv_multitaper_mean_fs-1_fmin-0_fmax-0-25 |
| plv_multitaper_mean_fs-1_fmin-0-25_fmax-0-5 | plv_multitaper_max_fs-1_fmin-0_fmax-0-5 | plv_multitaper_max_fs-1_fmin-0_fmax-0-25 | plv_multitaper_max_fs-1_fmin-0-25_fmax-0-5 |
| ppc_multitaper_mean_fs-1_fmin-0_fmax-0-5 | ppc_multitaper_mean_fs-1_fmin-0_fmax-0-25 | ppc_multitaper_mean_fs-1_fmin-0-25_fmax-0-5 | ppc_multitaper_max_fs-1_fmin-0_fmax-0-5 |
| ppc_multitaper_max_fs-1_fmin-0_fmax-0-25 | ppc_multitaper_max_fs-1_fmin-0-25_fmax-0-5 | pli_multitaper_mean_fs-1_fmin-0_fmax-0-5 | pli_multitaper_mean_fs-1_fmin-0_fmax-0-25 |
| pli_multitaper_mean_fs-1_fmin-0-25_fmax-0-5 | pli_multitaper_max_fs-1_fmin-0_fmax-0-5 | pli_multitaper_max_fs-1_fmin-0_fmax-0-25 | pli_multitaper_max_fs-1_fmin-0-25_fmax-0-5 |
| wpli_multitaper_mean_fs-1_fmin-0_fmax-0-5 | wpli_multitaper_mean_fs-1_fmin-0_fmax-0-25 | wpli_multitaper_mean_fs-1_fmin-0-25_fmax-0-5 | wpli_multitaper_max_fs-1_fmin-0_fmax-0-5 |
| wpli_multitaper_max_fs-1_fmin-0_fmax-0-25 | wpli_multitaper_max_fs-1_fmin-0-25_fmax-0-5 | dspli_multitaper_mean_fs-1_fmin-0_fmax-0-5 | dspli_multitaper_mean_fs-1_fmin-0_fmax-0-25 |
| dspli_multitaper_mean_fs-1_fmin-0-25_fmax-0-5 | dspli_multitaper_max_fs-1_fmin-0_fmax-0-5 | dspli_multitaper_max_fs-1_fmin-0_fmax-0-25 | dspli_multitaper_max_fs-1_fmin-0-25_fmax-0-5 |
| dswpli_multitaper_mean_fs-1_fmin-0_fmax-0-5 | dswpli_multitaper_mean_fs-1_fmin-0_fmax-0-25 | dswpli_multitaper_mean_fs-1_fmin-0-25_fmax-0-5 | dswpli_multitaper_max_fs-1_fmin-0_fmax-0-5 |
| dswpli_multitaper_max_fs-1_fmin-0_fmax-0-25 | dswpli_multitaper_max_fs-1_fmin-0-25_fmax-0-5 | dtf_multitaper_mean_fs-1_fmin-0_fmax-0-5 | dtf_multitaper_mean_fs-1_fmin-0_fmax-0-25 |
| dtf_multitaper_mean_fs-1_fmin-0-25_fmax-0-5 | dtf_multitaper_max_fs-1_fmin-0_fmax-0-5 | dtf_multitaper_max_fs-1_fmin-0_fmax-0-25 | dtf_multitaper_max_fs-1_fmin-0-25_fmax-0-5 |
| ddtf_multitaper_mean_fs-1_fmin-0_fmax-0-5 | ddtf_multitaper_mean_fs-1_fmin-0_fmax-0-25 | ddtf_multitaper_mean_fs-1_fmin-0-25_fmax-0-5 | ddtf_multitaper_max_fs-1_fmin-0_fmax-0-5 |
| ddtf_multitaper_max_fs-1_fmin-0_fmax-0-25 | ddtf_multitaper_max_fs-1_fmin-0-25_fmax-0-5 | dcoh_multitaper_mean_fs-1_fmin-0_fmax-0-5 | dcoh_multitaper_mean_fs-1_fmin-0_fmax-0-25 |
| dcoh_multitaper_mean_fs-1_fmin-0-25_fmax-0-5 | dcoh_multitaper_max_fs-1_fmin-0_fmax-0-5 | dcoh_multitaper_max_fs-1_fmin-0_fmax-0-25 | dcoh_multitaper_max_fs-1_fmin-0-25_fmax-0-5 |
| pdcoh_multitaper_mean_fs-1_fmin-0_fmax-0-5 | pdcoh_multitaper_mean_fs-1_fmin-0_fmax-0-25 | pdcoh_multitaper_mean_fs-1_fmin-0-25_fmax-0-5 | pdcoh_multitaper_max_fs-1_fmin-0_fmax-0-5 |
| pdcoh_multitaper_max_fs-1_fmin-0_fmax-0-25 | pdcoh_multitaper_max_fs-1_fmin-0-25_fmax-0-5 | gpdcoh_multitaper_mean_fs-1_fmin-0_fmax-0-5 | gpdcoh_multitaper_mean_fs-1_fmin-0_fmax-0-25 |
| gpdcoh_multitaper_mean_fs-1_fmin-0-25_fmax-0-5 | gpdcoh_multitaper_max_fs-1_fmin-0_fmax-0-5 | gpdcoh_multitaper_max_fs-1_fmin-0_fmax-0-25 | gpdcoh_multitaper_max_fs-1_fmin-0-25_fmax-0-5 |
| sgc_nonparametric_mean_fs-1_fmin-0_fmax-0-5 | sgc_nonparametric_mean_fs-1_fmin-0_fmax-0-25 | sgc_nonparametric_mean_fs-1_fmin-0-25_fmax-0-5 | sgc_nonparametric_max_fs-1_fmin-0_fmax-0-5 |
| sgc_nonparametric_max_fs-1_fmin-0_fmax-0-25 | sgc_nonparametric_max_fs-1_fmin-0-25_fmax-0-5 | sgc_parametric_mean_fs-1_fmin-0_fmax-0-5_order-None | sgc_parametric_mean_fs-1_fmin-0_fmax-0-25_order-None |
| sgc_parametric_mean_fs-1_fmin-0-25_fmax-0-5_order-None | sgc_parametric_mean_fs-1_fmin-0_fmax-0-5_order-1 | sgc_parametric_mean_fs-1_fmin-0_fmax-0-25_order-1 | sgc_parametric_mean_fs-1_fmin-0-25_fmax-0-5_order-1 |
| sgc_parametric_mean_fs-1_fmin-0_fmax-0-5_order-20 | sgc_parametric_mean_fs-1_fmin-0_fmax-0-25_order-20 | sgc_parametric_mean_fs-1_fmin-0-25_fmax-0-5_order-20 | sgc_parametric_max_fs-1_fmin-0_fmax-0-5_order-None |
| sgc_parametric_max_fs-1_fmin-0_fmax-0-25_order-None | sgc_parametric_max_fs-1_fmin-0-25_fmax-0-5_order-None | sgc_parametric_max_fs-1_fmin-0_fmax-0-5_order-1 | sgc_parametric_max_fs-1_fmin-0_fmax-0-25_order-1 |
| sgc_parametric_max_fs-1_fmin-0-25_fmax-0-5_order-1 | sgc_parametric_max_fs-1_fmin-0_fmax-0-5_order-20 | sgc_parametric_max_fs-1_fmin-0_fmax-0-25_order-20 | sgc_parametric_max_fs-1_fmin-0-25_fmax-0-5_order-20 |
| lmfit_SGDRegressor | lmfit_Ridge | lmfit_Lasso | lmfit_ElasticNet |
| lmfit_BayesianRidge | gpfit_DotProduct | gpfit_RBF | coint_johansen_max_eig_stat_order-0_ardiff-10 |
| coint_johansen_trace_stat_order-0_ardiff-10 | coint_johansen_max_eig_stat_order-0_ardiff-1 | coint_johansen_max_eig_stat_order-1_ardiff-1 | coint_johansen_max_eig_stat_order-1_ardiff-10 |
| coint_johansen_trace_stat_order-1_ardiff-10 | coint_johansen_trace_stat_order-0_ardiff-1 | coint_johansen_trace_stat_order-1_ardiff-1 | coint_aeg_tstat_trend-c_autolag-aic_maxlag-10 |
| coint_aeg_tstat_trend-ct_autolag-aic_maxlag-10 | coint_aeg_tstat_trend-ct_autolag-bic_maxlag-10 | pec | pec_orth |
| pec_log | pec_orth_log | pec_orth_abs | pec_orth_log_abs |

**Significance.** Within each dataset, we quantify the MDD–HC separation using the Mann–Whitney $U$ statistic, a rank-based nonparametric test that does not assume Gaussianity (McKnight & Najab, 2010). We apply the same testing protocol and decision threshold uniformly across all pipeline–SPI configurations.

**Consistency.** Because both datasets are modest in size, we use the nonparametric bootstrap to approximate the sampling distribution of our pipeline-level statistics and to assess directional stability under resampling (DiCiccio & Efron, 1996). Concretely, we treat the effect direction as supported when resampled estimates concentrate on the same sign (equivalently, when a bootstrap confidence interval for the contrast excludes zero), and we use this stability signal for robustness analysis and feature selection.

### C.4. Experiment Design

We partitioned the subjects into four groups based on dataset (MODMA vs. CISIR) and diagnosis (MDD vs. HC). For each SPI and each group, we computed individual-level rsFC asymmetry matrices and then averaged them within the group to obtain a single group-mean asymmetry matrix per SPI. We then computed the Spearman correlation between each pair of SPI representations to check the pairwise relationships. To enhance visualization, we performed hierarchical clustering on the MODMA-MDD subset and applied the resulting order across all four groups. Following previous work (Cliff et al., 2023), we assessed the relationship between SPIs by computing the pairwise Spearman correlation of their group-level rsFC asymmetry matrices.

### C.5. Environment

Experiments are performed on an 8-GPU (H20) high-performance computing cluster provided by the Large-scale Instrument Sharing Platform of Southwest University.

### C.6. Results

**Hidden hypothesis space (C1).** *Figure 5*a shows that the observation model is not a minor analysis choice but a defining part of the data interface. The block-structured correlation matrix indicates that equally defensible SPIs induce distinct, only weakly interchangeable representations of connectivity from the same measurements, consistent with a fragmented effective hypothesis space under different pipelines. Notably, `xcorr` (abbreviated from `xcorr_sq_mean_sig=False`) does not align with the dominant clusters, suggesting that selection heuristics based on consensus or similarity can systematically exclude pipelines that encode different mechanistic hypotheses. This supports the **(C1)** claim that the released representation does not uniquely specify the observation model that produced it, nor the conditions under which it should be trusted.

**Uncertified transportability (C2).** *Figure 5*c illustrates how a pipeline can fail to transport when the measurement regime changes. Compared with `xcorr`, `bary-sq` shows coherent structure in MODMA but degrades into unstructured patterns in CISIR, while `ppc` produces patterns consistent with an ill-suited pipeline for this target. These contrasts operationalize **(C2)**: even when a pipeline is plausible and can yield strong within-dataset signals, its regime of validity is not guaranteed, and failures under shift cannot be adjudicated without reporting transport conditions. Together, panels a to c show that observation model choice is a primary source of hypothesis space variation, transport failure, and unpropagated uncertainty in this audit.

**Ungoverned multiplicity (C3).** *Figure 5*b plots each SPI by within-dataset significance and cross-dataset directional consistency. The dense region of pipelines with high significance but low consistency illustrates that many defensible pipelines can produce apparently significant findings in a single dataset while failing to preserve the conclusion across datasets. Thus, without an explicit governance rule, researchers can defensibly select a significant pipeline and report a finding that does not survive admissible variation and setting shifts. Only one configuration (`xcorr`, red) satisfies the joint criteria in both datasets, matching the near zero survival rate reported in the main paper.

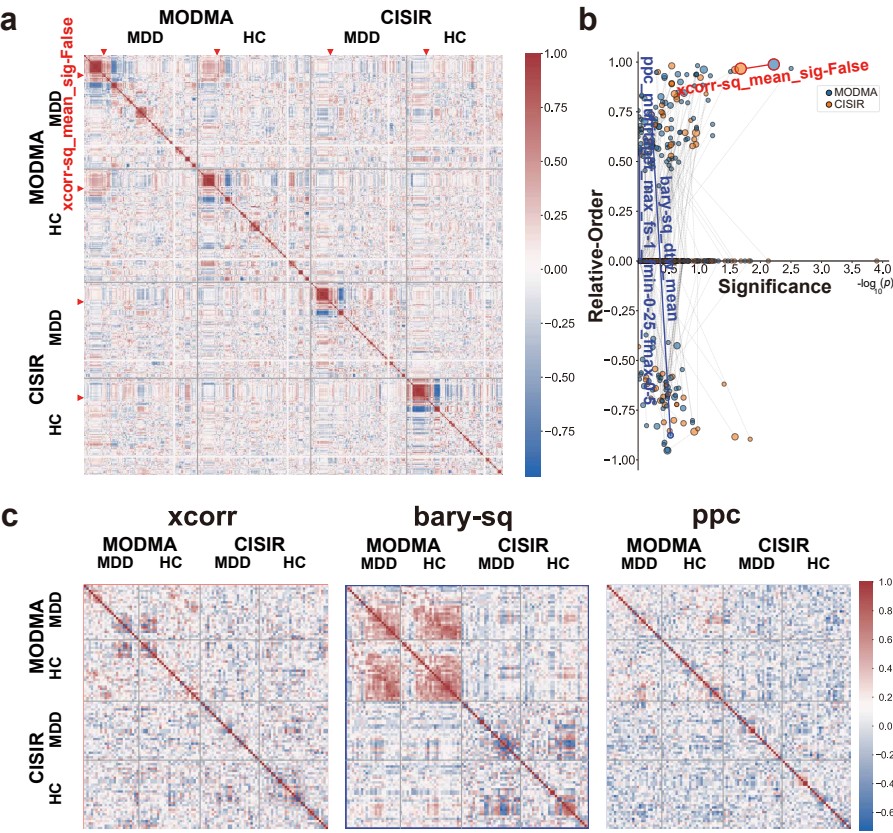

*Figure 5.* **Empirical demonstration of frozen lens failure modes (C1 to C3) in functional connectomics. (a) Hidden hypothesis space (C1).** Pairwise correlations among 284 candidate observation models (SPIs) show strong block structure, indicating that different defensible pipelines induce qualitatively different representations from the same measurements. **(b) Ungoverned multiplicity (C3).** The SPI landscape plotted by within-dataset significance and cross-dataset directional consistency shows that many pipelines look significant in isolation but fail to preserve conclusions across datasets. Only one configuration (xcorr, red) satisfies the joint criteria. **(c) Uncertified transportability (C2).** Qualitative comparison of the robust xcorr pipeline and unstable alternatives across datasets. bary-sq produces coherent structure in MODMA that collapses in CISIR, illustrating that a pipeline can appear valid in one measurement regime yet fail under a shifted regime.

