# OpenReview forum: "Position: AI for Science Should Treat Measurement-to-Dataset Pipelines as Inference Components"
_ICML.cc/2026/Position_Paper_Track — ICML 2026 Position Paper Track regular_

### Official Review · Reviewer_9azp · 2026-02-23

**Significance:** 4
**Argument Clarity:** 3
**Rating:** 5
**Confidence:** 4

**Questions:**

See the weakness section

**Alternative Views Section:**

Yes

**Compliance With Llm Reviewing Policy A Conservative:**

Affirmed.

**Discussion Potential:**

4

**Final Justification:**

I believe this paper makes a solid contribution. I keep my recommendation for Accept.

**Paper Summary:**

The paper argues that the observation-to-conclusion pipeline in AI4Science has a **frozen-lens problem**. The problem occurs when downstream inspection is conducted with a frozen upstream observation model ($P_\phi$). Though the frozen observation model is often a result of pragmatic choice, it is **not epistemically neutral**. So **it should be governed rather than treated as a default**. The argument advances along three diagnoses:

- **C1 hidden hypothesis space**: the pipeline determined what effects can be possibly discovered. Thus, the effective space of discoveries is pipeline-conditional.
- **C2 uncertified transportability**: learners optimized towards a fixed surrogate remains brittle under admissible shifts in the pipeline.
- **C3 ungoverned multiplicity**: cross-study synthesis can become ill-posed, not just noisy, because ungoverned selection of pipelines does not appropriately propagate uncertainty.

The paper calls to actions for **empirical audit** to stress-test frozen-lens interfaces. C1-C3 defines an empirical landscape. Pipeline selection should be an informed choice within such landscape in light of a computational budget, rather than a conventional default.

Finally, an agenda is proposed for establishing **computable observation frameworks**. what AI4Science lacks is not another checklist, but a way to make observation-model variation a computable object.

**Position:**

Yes

**Position In Title:**

Yes

**Related Work:**

3

**Strengths And Weaknesses:**

### Strengths

1. **Illuminating view**: pipeline choices determine an empirical landscape so downstream scientific claims are never pipeline-neutral, but are tied to a specific coordinate in the empirical landscape.
2. **Includes a case study**: the case study of EEG illustrates what the proposed auditing method looks like and greatly enhances the **actionability** this paper.
3. **Thoughtful response to alternative views**: $\S5.2$ is particularly punchy. The combinatorial space of degree-of-freedoms to be audited is not a problem to start with, but a result of ungoverned pipeline selection. When pipeline selection enables computable governance, auditing will become practical.

### Weaknesses

They are not necessarily weaknesses. Just suggestions in light of my standard ML background. So you can treat it as a point of reference for when you might lose reader coming from standard ML.

**1. The narrative of page2-3 can be improved** (I was kind of lost at page2-3)

**1.1 Provide a concrete example for “AI4Science — Indirect Observation”**

Figure1 has a lot going on. It’s unclear which arrows correspond to training an AI model versus actions by experimental scientists. I’m familiar with the standard ML pipeline but honestly my imagination of an AI4Science pipeline is very blurry. When you say “there are many degrees of freedom corresponding to ad hoc choices”, I believe it is very true and indeed reveals a serious problem, but cannot imagine what those degrees of freedom are. It could be better if you provide a parallel figure with all blocks replaced by concrete examples. You could use the EEG pipeline as an example since it will show up again in your case study. In EEG, the pipeline spans from a brain scanner to some scientific claim, then what do $z, \psi, P_\phi, f, x, \hat{y}$ correspond to?

**1.2 Illuminate the shift from pre-AI to post-AI in the “Evolution of the Frozen Lens”**

It is unclear to me what roles foundational AI models play in the frozen-lens problem. Did AI create new problems? Did AI exacerbate the degree of existing problems? Did AI add new dimensions to existing problems? I assume that prototypes of C1, C2, C3 existed in pre-AI era (i.e. the classical paradigm in Table1). What auditing and governance approaches had been attempted? Were they rendered inadequate in post-AI era? I believe the frozen-lens problem is persistent throughout the history of natural science and evolves continuously. Thus, providing a richer historical story can be greatly helpful.

**2. The relationship among actionable items in $\S6$ can be elaborated**

I’m curious whether there are conditions among $\S6.1-6.4$. For example, should the four steps be achieved sequentially because the previous step serves as a prerequisite for the next? Or can they be initially pursued in parallel and the north-star goal is the maturation of all four? Will insufficiency in one step obscure the accomplishment of another step?

**3. Map call-to-actions to a division-of-expertise**

$\S5.3$ states that developing a computable observation framework is a shared responsibility among natural scientists, ML scientists, and governing parties. It would be great to sort actionable items in $\S6$ according to the type of expertise demanded (maybe a new table is needed here). For example, as an ML scientist, I don’t know what degrees-of-freedom are for a particular scientific practice. So even though I agree with your proposals, I don’t know what I can do. However, “DAG as an export target” makes me excited because I feel like the next DAG-to-$\langle$some output$\rangle$ step is within my expertise.

**Support:**

4

---

> ### Author Rebuttal · Authors · 2026-03-29
>
> Thank you for the thoughtful and constructive review. We especially appreciate your **strong assessment of the paper’s support, significance, and discussion potential**. Your comments show clearly where the presentation can be made more concrete and more accessible to readers coming from a standard ML background.
>
> We will revise accordingly.
>
> **W1.1 Provide a concrete example for "AI4Science"**
> We agree that the pages 2--3 narrative and Figure 1 are currently too abstract. In revision, we will clarify which parts of Figure 1 correspond to data generation, preprocessing, and downstream learning, and add a parallel example based on the EEG case study. In that example, $x$ will denote raw EEG recordings; $P_{\phi}$ the preprocessing/reconstruction pipeline, where $\phi$ indexes choices such as filtering, artifact removal, referencing, epoching, and connectivity construction; $z=P_{\phi}(x)$ the released representation; $f$ the downstream learner on $z$; $\hat{y}$ the proxy-task output; and $\psi$ the latent scientific mechanism inferred only indirectly. We will use this mapping to show concretely that the relevant “degrees of freedom” are admissible choices in processing and representation.
>
> **W1.2 Illuminate the shift from pre-AI to post-AI**
> Our intent is not to suggest that the frozen-lens problem began with AI. Prototypes of C1--C3 existed in the pre-AI era: indirect-observation sciences long relied on convention-driven preprocessing pipelines, with governance via standard toolchains, reporting norms, benchmark datasets, and best-practice checklists. These approaches stabilize procedures and defaults, but they do not make admissible pipeline variation an explicit, computable object that can be searched, stress-tested, and propagated into claims. AI therefore primarily **amplifies** an existing problem, while also adding new dimensions through scale, opacity, reuse, and automation. As AI4Science shifts toward reusable derivatives, benchmark-driven optimization, foundation-model reuse, agents, and automated experimentation, frozen upstream interfaces travel farther, persist longer, and are optimized more aggressively. In revision, we will make this continuity-and-escalation story clearer and explain more directly why post-AI workflows expose the limits of checklist-style governance.
>
> **W2. The relationship among actionable items in Sec. 6**
> We agree that the current presentation makes Sections 6.1--6.4 seem too flat. Our intended view is neither fully sequential nor fully independent. Section 6.1 (representation) is the main enabling layer: without a typed, executable pipeline representation, later optimization and audit/governance remain bespoke and non-cumulative. Once some representation exists, 6.2 and 6.3 can advance in parallel: optimization can search for stability-aligned pipelines over a partial admissible space, while audit/governance can standardize reporting, reproduce results across datasets, and accumulate stability profiles. The goal is maturation of all layers together, but later layers need not wait for earlier ones to be complete. The layers are also not independent: without 6.1, 6.2 and 6.3 are hard to scale or reuse. Without 6.2, 6.3 can diagnose instability but cannot support principled selection or aggregation. Without 6.3, gains in representation and optimization remain local and do not accumulate into governed cross-study evidence. We will revise Sec. 6 to make this structure explicit and clarify that 6.4 is the closing-the-loop integration of the earlier layers rather than a separate fourth step.
>
> **W3. Map the call to action to a division of expertise**
> We strongly agree that the paper should make clearer what different communities can do. We will add a table or figure in Sec. 6 distinguishing three roles: (1) **domain scientists**, who define admissible degrees of freedom, scientific validity conditions, and which forms of variation count as meaningful or artifactual; (2) **ML researchers**, who contribute representations, search procedures, stability-aligned objectives, uncertainty-aware selection/aggregation rules, and export targets such as DAG-based pipeline representations; and (3) **interdisciplinary methodologists**, who support executable standards, audit interfaces, release requirements, and cumulative reporting across labs and datasets. We will also revise Sec. 6 so that ML-facing entry points are more explicit: once domain experts specify admissible pipeline choices, there is a substantial ML research agenda around formalizing those choices as structured search spaces, optimizing over mixed discrete/continuous pipeline families under compute constraints, defining stability-based selection criteria across datasets, and quantifying pipeline uncertainty in downstream evaluation.
>
> By making these entry points explicit, we hope the paper clarifies that this is not only a domain-science standards question, but also a concrete ML research agenda for the broader ICML audience.

---

> > ### Author Rebuttal · Reviewer_9azp · 2026-04-01
> >
> > **W1.2** "AI therefore primarily **amplifies** an existing problem" --- I agree that this is a precise characterization
> >
> > **W2** 6.1 is the main enabling layer, 6.2 and 6.3 can be down in parallel, 6.4 is the closing-the-loop integration. I look forward to seeing a flowchart-like figure in the revision.
> >
> > **W3** I like the expertise-aware call-to-action

---

> > > ### Author Response · Authors · 2026-04-06
> > >
> > > Thank you for your thoughtful review and for the time and effort you devoted to evaluating our paper. We are very glad that our clarifications were helpful. We will revise the paper accordingly, and we believe the rebuttal process has significantly improved the quality and clarity of the paper.

---

### Official Review · Reviewer_6LAe · 2026-03-10

**Significance:** 2
**Argument Clarity:** 3
**Rating:** 5
**Confidence:** 3

**Questions:**

- Can you explain in what sense an invertible representation (IR) is “invertible”?
- In the EEG experiment, I didn’t fully get what the 284 operators (statistical pairwise interactions) are exactly? Can you elaborate or point me to the place (in the appendix?) where this is explained?
- In Figure 1, how is y (or yhat) related to z and x?
- In Figure 1, does “learner” and “ML model” mean the same thing? Might just be that you chose to represent the ML4Science pipeline in more detail.

**Alternative Views Section:**

Yes

**Compliance With Llm Reviewing Policy A Conservative:**

Affirmed.

**Discussion Potential:**

4

**Final Justification:**

Thank your for your clarifying responses. I increased my score to 5.

**Paper Summary:**

This position paper argues that AI4Science methods should not just work on a single derived dataset (after post processing) but rather include the whole pipeline that leads to the derived dataset. It is a clear position paper that, I think, satisfies all the requirements of a position paper.  The position seems closely related to multiverse analysis (as the authors note themselves) but with some twists, which relate to the explicit representation of preprocessing pipelines as optimizable objects. I took me a bit to time to get into the terminology of the paper, but once understood, I think the terminology and thought processes make a lot of sense.

**Position:**

Yes

**Position In Title:**

Yes

**Related Work:**

3

**Strengths And Weaknesses:**

Strengths:

- The paper has a clear and defensible position, clearly argued for and supported by evidence

Weaknesses:

- The paper uses a lot of jargon at times, which makes it not super accessible to a wider ML audience I believe. Despite the content and claims being, in principle, easily understandable by a wide audience if presented somewhat less technical (at least in the early parts of the paper).
- Once the main point is understood, I think the content across section 1-3 become somewhat repetitive. Not sure if this is a weakness of just the nature of these kind of position papers really.
- The presented empirical example (Section 4) is especially hard to understand because it uses a lot of EEG specific abbreviations and terminology that is used but not explained, or insufficiently so (e.g., in Table 2). Also the pipeline seems to just use a bunch of simple statistical tests in the end, making it look like a basic multiverse analysis. So I am not super convinced how this examples provides specific enough evidence for the concrete position being taken.
- A lot of choices in the pre-processing pipeline may be discrete rather than continuous. The authors talk about gradient-based tuning in Section 6, which would largely be at odds with a large discrete design space of many pipelines.

**Support:**

3

---

> ### Author Rebuttal · Authors · 2026-03-29
>
> Thank you for the careful and constructive review. We appreciate that you found the paper to have a **clear and defensible position**, **supported by evidence**, and especially that you viewed it as having **strong discussion potential**. We also appreciate your point that, once the terminology is understood, the core logic becomes much easier to follow.  Your comments show clearly where the presentation can be made more accessible to a broader ML audience.
>
> We will revise accordingly.
>
> **W1. Early abstraction, jargon, and repeated framing across Sections 1--3.**
> We agree that, especially early in the paper, some of the interdisciplinary terminology can obscure what is ultimately a fairly simple core ML argument. In revision, we will streamline the C1--C3 exposition, reduce repeated framing across Sections 1--3, and translate domain-specific terminology into more standard ML language so that the position is easier to enter for a broader ICML audience.
>
> **W2. EEG audit and multiverse-style analysis.**
> We agree that Section 4 can reasonably be read as *multiverse-style* in mechanics, because it enumerates a large set of admissible preprocessing choices and examines how downstream results vary across them. The distinction is not the mechanics of the sweep but its inferential target.
>
> Standard multiverse analyses often ask whether a finding remains robust across reasonable analytic choices. In our case, the EEG audit uses a multiverse-style sweep for a different purpose: to stress-test whether the released data interface is a stable evidential object at all. The goal is not to verify a biomarker, but to show that the released data interface can vary enough across pipelines to change what appears stable, generalizable, or scientifically meaningful. That is why we argue that preprocessing should be treated as an explicit inference component rather than being fixed implicitly by defaults, convention, or authority.
>
> In revision, we will make this logic explicit at the start of Section 4, reduce EEG-specific abbreviations, define the terminology needed to read Table 2 in plain language, and clarify why this audit bears on the paper’s concrete position rather than serving only as a generic multiverse exercise.
>
> **W3. Discrete pipeline spaces and Section 6.**
> We agree that many preprocessing choices are inherently discrete, so a gradient-only framing would be too narrow. Our intended claim in Section 6 is broader: the core requirement is *computability*, not differentiability. Our goal is for pipeline representations to support not only gradient-based tuning where appropriate, but also budgeted gradient-free search and policy-based selection over admissible pipeline spaces. We agree, however, that the current wording gives gradient-based tuning too much prominence and can therefore suggest the wrong optimization picture. In revision, we will rewrite Section 6 to make this intended claim explicit.
>
> **Q1. IR / “invertible.”**
> By “invertible,” we mean that the pipeline representation should support bidirectional traceability between specification and execution: a machine-readable specification can be compiled into an executable pipeline, and the executed pipeline can be decoded, audited, and replayed under declared admissible choices. We agree that “invertible” is too easy to misread. In revision, we will replace it with language emphasizing executable, machine-readable pipeline specifications and bidirectional traceability.
>
> **Q2. “284 operators” / SPIs.**
> These are the 284 Statistical Pairwise Interactions (SPIs) used as downstream connectivity operators applied to the preprocessed EEG signal. More concretely, SPIs are algorithms for measuring interactions between pairs of time series as real-valued summary statistics. Standard measures such as Pearson correlation are canonical examples within this family. In revision, we will define SPIs explicitly at first mention (Section 4.1) and add an appendix table in Appendix C.3 summarizing the SPI families.
>
> **Q3. Figure 1: relation among $x$, $z$, and $\hat{y}$; “learner” vs. “ML model.”**
> We agree that Figure 1 currently leaves the dependencies across the two panels too implicit. In the Standard ML panel, the model operates directly on the dataset and produces the prediction. In the AI4Science panel, raw measurements $x$ are first transformed into the released representation $z=P_{\phi}(x)$, and the learner then produces the proxy-task prediction $\hat{y}=f_{\theta}(z)$. The intended contrast is that the learner only accesses the pipeline-induced representation $z$, not the underlying system directly. “Learner” and “ML model” are intended to play the same functional role here. We agree that using both terms creates unnecessary ambiguity. In revision, we will make this flow explicit in the figure and caption, unify the terminology, and add a concrete EEG walk-through showing what $x$, $P_{\phi}$, $z$, $f_{\theta}$, $\hat{y}$, and $\psi$ correspond to in practice.

---

> > ### Author Rebuttal · Reviewer_6LAe · 2026-04-01
> >
> > Thank your for your clarifying responses.

---

> > > ### Author Response · Authors · 2026-04-06
> > >
> > > Thank you for your careful review and for the time and effort you devoted to evaluating our paper. We are very glad to know that our rebuttal adequately addressed your concerns. We believe the rebuttal process has significantly improved the quality and clarity of the paper. We sincerely appreciate your constructive feedback throughout the review process.

---

### Official Review · Reviewer_nwTb · 2026-03-11

**Significance:** 3
**Argument Clarity:** 4
**Rating:** 5
**Confidence:** 4

**Questions:**

I have two questions addressing the weaknesses described:
1. Would the authors reconsider the dichotomy they present in favour of a degree of importance of the pipeline step, and also that problems outside AI4Science may benefit from this thinking?
2. Would the authors consider the biological data mentioned as relevant to the arguments they present?

**Alternative Views Section:**

Yes

**Compliance With Llm Reviewing Policy A Conservative:**

Affirmed.

**Discussion Potential:**

4

**Paper Summary:**

This paper argues that, specifically in the AI4Science problem space, the processing pipeline from raw sensor readings, and the conditions under which these readings are taken, should form a key part of the modeling process in AI4Science problems.

The core argument is that current datasets are produced by a variety of processing pipelines, each calibrated differently, that condense experimental readings into a single interpretation of experimental data, ignoring experimental uncertainty. Consequently the datasets can comprise of incongruous data that together cannot be interpreted properly, despite the fact that the underlying experimental data may be well explainable on its own, or had the data all been processed in under a single pipeline.

This is demonstrated on an EEG measurement example with a dataset of observations from two sources processed with numerous data pipelines from the literature.

The call to action is to document well the pipelines used to generate data, to allow for gradient based optimisation through these pipelines, and to centralise the collection of this data.

**Position:**

Yes

**Position In Title:**

Yes

**Related Work:**

3

**Strengths And Weaknesses:**

I find this position compelling and an important consideration in discussion of fitting machine learning to experimental data from multiple sources. The examples given are clear and cover a wide range of scientific topics.

I would point out two things I think could be improved.

Firstly, I feel the authors are forcing a somewhat false dicotamy onto AI for science tasks and other ML tasks. The AI4Science layout in figure 1 could well be applied for example to image data - there is a required processing pipeline to turn raw image sensor data into pixel images that fits the description well. We generally trust this pipeline, but for example varying exposure, and how we interpret the mixture of signals in a pixel to produce a color, are subject to a large amount of processing. My point is not to disagree with the premise, that the processing pipeline is important, but that this is not exclusive to AI4Science, nor is it a dichotomy, but a question of to what degree the pipeline has slack to interpret experimental data.

Secondly I was surprised to not see structural biological applications discussed. The false homogeneity for example of the Protein Data Bank is a prime example of the many-pipelines data in one source, poorly annotated, and heavily reduced experimental error inclusion. On a related topic, the data collected in e.g. ChEMBL for biological interactions and used to train ML models is poorly annotated, and in much of the the experimental conditions under which a particular experiment was performed is missing.

**Support:**

3

---

> ### Author Rebuttal · Authors · 2026-03-29
>
> Thank you for the thoughtful and constructive review. We appreciate that you found the **position compelling**, the discussion **important**, and the examples **clear across a wide range of scientific topics**. We also appreciate your two suggestions, because they show exactly how the paper can become more precise and more useful to the ICML audience: (i) recasting the AI4Science-versus-standard-ML contrast so it is not read as a rigid dichotomy, while still clarifying why indirect-observation sciences often carry a distinct epistemic burden, and (ii) bringing structural biology examples more explicitly into scope.
>
> We will revise the paper accordingly.
>
> **W1. Recast the contrast as degree-based**
> Thank you for this important suggestion. We agree that the current framing can read as too dichotomous. Your comment helped us see that the more useful distinction is not whether a problem is “AI4Science,” but how much interpretive slack the upstream pipeline has before measurements become the released data interface, and what epistemic burden that slack places on downstream claims.
>
> In revision, we will therefore recast the contrast in degree-based terms. More concretely, we will clarify “degree of slack” using several practical considerations: how many admissible preprocessing choices exist, how strongly they affect the released representation or downstream result, how well they are documented or standardized, and how much of that variation remains visible at the released interface. Under this framing, pipeline slack is not exclusive to AI4Science. Even standard image pipelines can embed nontrivial upstream choices.
>
> Our intended claim, more precisely, is that in many **indirect-observation** settings the epistemic cost of that slack is different and often substantially higher. In many standard ML settings, the task is defined on an agreed interface, so preprocessing is mainly judged by whether it improves performance or robustness on that interface. In AI4Science under indirect observation, by contrast, the released representation is often a processed projection of measurements onto a latent system that is not directly observable. In that regime, the pipeline does not merely clean or format the input. It helps determine what counts as evidence for the underlying mechanism, and thus whether downstream conclusions are mechanistically meaningful, transportable, or cumulative.
>
> We therefore do not mean to insist on a rigid boundary between AI4Science and the rest of ML. Rather, the paper identifies a family of settings in which upstream measurement-to-dataset choices have enough interpretive slack, enough irreversibility, and enough impact on scientific claims that they should be treated as first-class inference components. We will revise the framing around Figure 1 and the surrounding text accordingly.
>
> **W2. Bring structural biology examples more explicitly into scope**
> Thank you for this very helpful suggestion. We agree that the structural biology examples you mention are highly relevant to the paper’s argument and well within our intended scope. The current manuscript already argues that the frozen-lens problem is structural under indirect observation rather than specific to EEG, but we agree that examples such as Protein Data Bank aggregation and ChEMBL-style assay/condition heterogeneity would make that broader relevance more concrete to additional communities [1,2].
>
> We will therefore revise the paper to incorporate this class of examples explicitly. We will show that they instantiate the same broader pattern: community-facing interfaces can present a false homogeneity over upstream choices, experimental conditions, reconstruction or curation conventions, or missing error annotations, with the result that downstream users inherit a frozen observation model without being able to fully inspect, vary, or propagate that uncertainty. Concretely, we will expand the Pipeline Exemplars in **Appendix A** and revise the corresponding introduction so that structural-biology cases appear more explicitly as additional instances of the same general issue.
>
> More broadly, your review helps us sharpen an important point: the paper is not arguing that only AI4Science has pipeline slack. It is arguing that indirect-observation settings make upstream pipeline commitments especially consequential, because the released interface is a model-laden representation of an underlying system. That is the regime in which preprocessing pipelines should be treated as inference components rather than as fixed dataset definitions.
>
> [1] Lam J H, Katritch V. Navigating structure-based drug discovery with emerging innovations in physics- and knowledge-based approaches[J]. *npj Drug Discovery*, 2025, 2(1): 29.
>
> [2] Papadatos G, Gaulton A, Hersey A, et al. Activity, assay and target data curation and quality in the ChEMBL database[J]. *Journal of Computer-Aided Molecular Design*, 2015, 29(9): 885-896.

---

> > ### Author Rebuttal · Reviewer_nwTb · 2026-04-07
> >
> > Thanks for the rebuttal.
> > W1 - great I think this is a clearer framing of the problem and an easy to grasp one.
> > W2 - sounds like a good improvement to me, thank you.
> > I will keep my score as is but I'm glad if this has helped improve the paper.

---

> > > ### Author Response · Authors · 2026-04-07
> > >
> > > Thank you for your thoughtful acknowledgement and for your constructive suggestions. We are glad that the revised framing and broader scope were helpful. We will incorporate these improvements in the final version to further strengthen the paper’s clarity and generality, making it more actionable for the broader ICML community.

---

### Official Review · Reviewer_kSEH · 2026-03-13

**Significance:** 3
**Argument Clarity:** 3
**Rating:** 5
**Confidence:** 4

**Questions:**

see weaknesses.

**Alternative Views Section:**

Yes

**Compliance With Llm Reviewing Policy A Conservative:**

Affirmed.

**Discussion Potential:**

3

**Paper Summary:**

Standard ML treats datasets as fixed, given as text or pixels. However, AI4Science datasets are constructed using indirect observations typically. For instance, MR images are generated by collecting Fourier space data and then solving an inverse problem.

This paper posits that AI4Science workflows should treat measurement-to-dataset pipelines as inference components, therefore pipelines should be audited, evaluated instead of being treated as frozen. The authors claim this creates three failure modes:

1. **Hidden hypothesis space:** the dataset does not fully specify the pipeline configuration that generated it.
2. **Uncertified transportability:** pipeline validity may break under distribution shift.
3. **Ungoverned multiplicity:** multiple defensible pipelines exist but their variability is not propagated into uncertainty estimates.

The authors then make arguments in support of why such audits can improve over current practices and prevent erroneous inferences, backed with empirical evidence. The authors show the effect of these pipeline choices on inferences drawn from the datasets and how not taking into the pipeline can lead to erroneous conclusions.

The authors also consider relevant alternate views and present convincing arguments against these. Overall, the paper presents a relevant call to action for the AI4Science community in the form of the computable observation framework.

**Position:**

Yes

**Position In Title:**

Yes

**Related Work:**

3

**Strengths And Weaknesses:**

The paper highlights an important methodological issue in AI4Science, datasets derived from indirect observations depend heavily on preprocessing pipelines, yet these pipelines are typically treated as fixed inputs rather than inference components.

AI for healthcare particularly suffers from non-standard choices, for instance, different scanner types and  parameters, selection protocols, measurement units, etc.

The three proposed failure modes (hidden hypothesis space, uncertified transportability, ungoverned multiplicity) provide a useful conceptual taxonomy.

The empirical audit in EEG neuroscience is a compelling illustration of the problem. The large pipeline sweep (245,376 configurations) demonstrates how conclusions can vary dramatically under reasonable pipeline variation.

The proposed frameworks, however, remains largely conceptual. The paper does not present a concrete implementation or system demonstrating how such frameworks would be used in practice. However, the identification of this problem and standardizing communication and reporting is an important call to action. The position can be made stronger by including analysis on other datasets besides EEGs.

**Support:**

3

---

> ### Author Rebuttal · Authors · 2026-03-29
>
> Thank you for the careful and constructive review. We appreciate that you identified the paper as highlighting **an important methodological issue in AI4Science**, found the three failure modes a **useful conceptual taxonomy**, and viewed the EEG audit as a **compelling illustration** of how strongly downstream conclusions can vary under reasonable pipeline variation. We especially appreciate your broader healthcare framing: scanner heterogeneity, acquisition-parameter differences, protocol variation, and other non-standard measurement choices are exactly the kinds of settings in which frozen-pipeline assumptions become especially fragile. Your review makes clear how the paper can be strengthened further: by making the Computable Observation Framework (COF) feel less purely conceptual and more operational, and by showing more directly that the underlying phenomenon is not confined to the EEG case study.
>
> We agree and will revise the paper accordingly.
>
> **W1. Make the COF more operational**
> Thank you for this important point. We agree that the current draft can make the COF feel more conceptual than operational. In revision, we will present the framework as a minimal practical workflow rather than only as a high-level architecture. Concretely, this workflow has three parts:
> (1) an explicit, versioned specification of the admissible pipeline family;
>
> (2) a bounded audit that evaluates selected admissible variants across relevant datasets or settings and records claim stability or sensitivity; and
>
> (3) a reporting artifact that states what variation was explored, what stability has been established, and what residual pipeline uncertainty remains.
>
> We will also clarify that each of these components is already partly feasible in existing practice, even if they have not yet been assembled into the unified workflow proposed here. Bounded pipeline sweeps can already function as audits over admissible variation. Executable workflow descriptions show that explicit pipeline specification is feasible in principle. Current reporting practices, although incomplete, provide a starting point for communicating explored variation and residual uncertainty. In revision, we will connect these pieces more explicitly and present them as a minimal operational pathway for the COF.
>
> In that sense, our contribution is not to claim that each ingredient is new in isolation, but to show how executable pipeline specification, bounded audit over admissible variation, and structured reporting can be combined into a practical observation-model workflow. We will revise the paper to make this pathway more explicit and easier to act on.
>
> **W2. Show the breadth beyond EEG more directly**
> Thank you for this important point. We agree that the paper is stronger if the support is not perceived as resting on a single EEG example. The current manuscript already argues that the issue is broader than EEG and, in **Appendix A**, provides representative pipeline exemplars across four indirect-observation domains. We agree, however, that this breadth is not yet visible enough in the present framing.
>
> In revision, we will foreground this cross-domain grounding more explicitly in the introduction and connect the Section 4 EEG audit more directly to *existing evidence from resting-state fMRI*. Prior work has shown both that rs-fMRI connectomics is highly sensitive to upstream data-processing pipelines and that downstream functional connectivity organization varies substantially across connectivity-mapping methods [1,2]. We will use these results to clarify that the empirical dependence observed in EEG should be read not as a modality-specific anomaly, but as a concrete instance of a broader structural problem in indirectly observed sciences.
>
> Because this is a *position paper*, our primary goal is to clarify the community’s conceptual target rather than introduce a second benchmark-style empirical arc. In revision, we will therefore make the existing cross-domain support more explicit and connect the EEG audit more directly to analogous evidence in other indirect-observation settings, so that the argument’s generality is clearer without diffusing its core position.
>
> We appreciate these suggestions because they sharpen the paper in two concrete ways: they push us to present the COF as a minimal operational workflow, and to make the paper’s cross-domain scope more visible.
>
> [1] Luppi A I, Gellersen H M, Liu Z Q, et al. Systematic evaluation of fMRI data-processing pipelines for consistent functional connectomics[J]. Nature Communications, 2024, 15(1): 4745.
>
> [2] Liu Z Q, Luppi A I, Hansen J Y, et al. Benchmarking methods for mapping functional connectivity in the brain[J]. Nature Methods, 2025, 22(7): 1593-1602.

---

> > ### Author Rebuttal · Reviewer_kSEH · 2026-04-04
> >
> > Thank you for your response. Keeping my score!

---

> > > ### Author Response · Authors · 2026-04-06
> > >
> > > Thank you for your thoughtful review and for the time and effort you devoted to evaluating our paper. We are very glad to know that our rebuttal adequately addressed your concerns. We believe the rebuttal process has significantly improved the quality and clarity of the paper. We sincerely appreciate your constructive feedback throughout the review process.

---

### Decision · Program_Chairs · 2026-04-30

**Decision:**

Accept (regular)

**Comment:**

There was strong agreement amongst the reviewers, with particularly agreement on the large scale neuroscience audit provided as evidence and the general statement and explanation of the "frozen lens" problem.

From my own read, the paper would be improved by working in citations to "Uncertain Evidence in Probabilistic Models and Stochastic Simulators" and the earlier work of Jeffrey and Pearl cited within.  This line of work is highly relevant and should be known to the authors if it is not.